# OmniMouse: Scaling properties of multi-modal, multi-task Brain Models on 150B Neural Tokens

**Konstantin F. Willeke**[*,‡1,2,3,4], **Polina Turishcheva**[*4], **Alex Gilbert**[*1,2,3], **Goirik Chakrabarty**[4], **Hasan A. Bedel**[1,2,3], **Paul G. Fahey**[1,2,3], **Yongrong Qiu**[1,2,3], **Marissa A. Weis**[4], **Michaela Vystrčilová**[4], **Taliah Muhammad**[1,2,3], **Lydia Ntanavara**[1,2,3], **Rachel E Froebe**[1,2,3], **Kayla Ponder**[1,2,3], **Zheng Huan Tan**[1,2,3], **Emin Orhan**[6], **Erick Cobos**[7,8], **Sophia Sanborn**[1,2,3], **Katrin Franke**[1,2,3], **Fabian H. Sinz**[†4], **Alexander S. Ecker**[†4,5], **Andreas S. Tolias**[†1,2,3,7,8,9]

[1]Department of Ophthalmology, Byers Eye Institute, Stanford University
[2]Stanford Bio-X, Stanford University
[3]Wu Tsai Neurosciences Institute, Stanford University
[4]Institute of Computer Science and Campus Institute Data Science, University Göttingen
[5]Max Planck Institute for Dynamics and Self-Organization, Göttingen
[6]National Center for Computational Sciences, Oak Ridge National Laboratory
[7]Center for Neuroscience and Artificial Intelligence, Baylor College of Medicine
[8]Department of Neuroscience, Baylor College of Medicine
[9]Department of Electrical Engineering, Stanford University
[*,†]**Equal contribution**    [‡]**Corresponding author:** `willeke@stanford.edu`

## Abstract

Scaling data and artificial neural networks has transformed AI, driving breakthroughs in language and vision. Whether similar principles apply to modeling brain activity remains unclear. Here we leveraged a dataset of 3.1 million neurons from the visual cortex of 73 mice across 323 sessions, totaling more than 150 billion neural tokens recorded during natural movies, images and parametric stimuli, and behavior. We train multi-modal, multi-task models that support three regimes flexibly at test time: neural prediction, behavioral decoding, neural forecasting, or any combination of the three. OmniMouse achieves state-of-the-art performance, outperforming specialized baselines across nearly all evaluation regimes. We find that performance scales reliably with more data, but gains from increasing model size saturate. This inverts the standard AI scaling story: in language and computer vision, massive datasets make parameter scaling the primary driver of progress, whereas in brain modeling – even in the mouse visual cortex, a relatively simple system – models remain data-limited despite vast recordings. The observation of systematic scaling raises the possibility of phase transitions in neural modeling, where larger and richer datasets might unlock qualitatively new capabilities, paralleling the emergent properties seen in large language models. Code available at `https://github.com/enigma-brain/omnimouse`.

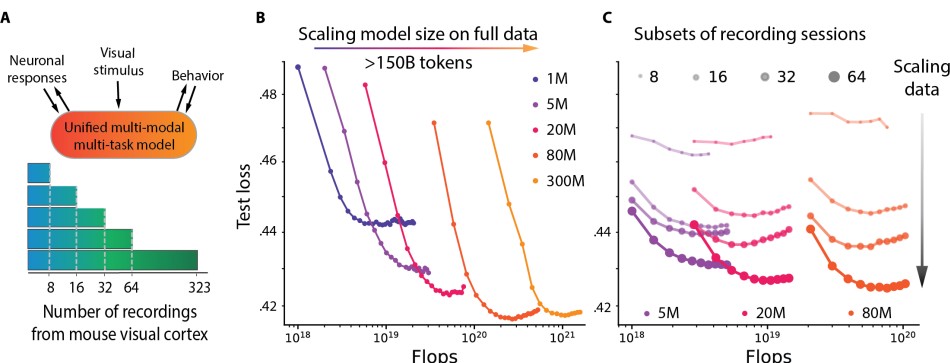

Figure 1: **A.** OmniMouse unifies neural prediction, behavior decoding, and forecasting tasks. **B.** Scaling model size on 150+ billion neural tokens shows performance saturation, unlike language models. **C.** In contrast, scaling data consistently improves performance across all model sizes.

# 1 INTRODUCTION

Scaling models and data has driven recent progress in machine learning, with large language, vision, and multi-modal models showing consistent performance gains and enabling foundation models that unify tasks across domains. A natural question is whether models of the brain can also benefit from scaling. The mouse visual cortex, with large-scale neural datasets (MICrONS Consortium et al., 2021; de Vries et al., 2019; Angelaki et al., 2025) and standardized benchmarks (Willeke et al., 2022; Turishcheva et al., 2024), offers a natural setting to investigate this, yet compared with the internet-scale corpora used in language and vision, available neural datasets are much smaller, more fragmented, and less diverse. The neuroscience community has recently started to work toward foundation models for EEG (Chau et al., 2024; Chen et al., 2024; Cui et al., 2024; Jiang et al., 2024; Kostas et al., 2021; Yang et al., 2023; Thapa et al., 2024; Li et al., 2024), fMRI (Caro et al., 2023; Dong et al., 2024; Kan et al., 2022; Thomas et al., 2022; d'Ascoli et al., 2025), MEG (Csaky et al., 2024), and intracranial signals (Zhang et al., 2023; Wang et al., 2023). Prior work in this direction focused on isolated modalities (Ye et al., 2023; Azabou et al., 2023), a single predictive task (Wang et al., 2025), lacked scalability across datasets (Ye & Pandarinath, 2021; Mi et al., 2023; Antoniades et al., 2024), or omitted stimulus (Zhang et al., 2025) and behavioral information (Jiang et al., 2025; Mi et al., 2023).

In this work, we introduce OmniMouse, a multi-modal, multi-task architecture for modeling neuronal activity in the mouse visual cortex. Unlike prior models that are typically restricted to a single modality, OmniMouse combines single-neuron tokenization, video encoding and behavioral decoding into a unified architecture. Our model design enables flexible masking, allowing the model to handle arbitrary combinations of neural forecasting, stimulus-conditioned response prediction, sub-population prediction, and behavioral decoding. We train OmniMouse on one of the largest single-neuron datasets to date: 323 recordings from the visual cortex of 73 awake mice viewing naturalistic movies, images, and parametric stimuli, totaling over 150 billion neuronal activity tokens from over 3 million neurons.

Our main findings and contributions are:

- **Systematic scaling analysis**: We find that performance improves systematically with more data, but saturates with model size beyond moderate scales. This suggests that data, not model size, is currently the bottleneck for predictive accuracy in neural modeling.
- **A multi-modal, multi-task model for visual cortex**: OmniMouse handles both single-modality and multi-modal inputs, supporting any combination of forecasting and stimulus-conditioned prediction across different neurons, stimuli, time points, and animals.
- **OmniMouse achieves state-of-the-art performance**: When compared to strong specialized baselines, OmniMouse outperforms prior specialized methods across nearly all tasks, showing that our architecture is competitive independent of data scale advantages.

# 2 RELATED WORK

**Large-scale deep learning models for single-neuron predictions.** Deep learning has advanced predictive modeling in neuroscience, particularly in vision (Cadieu et al., 2014; Batty et al., 2017; Klindt et al., 2017; McIntosh et al., 2016; Cadena et al., 2019; Kindel et al., 2019; Walker et al., 2019; Zhang et al., 2018; Ecker et al., 2018; Sinz et al., 2018; Burg et al., 2021; Cowley & Pillow, 2020). CNN-based approaches introduced shared feature cores with per-neuron readouts (Antolík et al., 2016; Klindt et al., 2017; McIntosh et al., 2016; Sinz et al., 2018; Lurz et al., 2021), culminating in multi-animal "digital twins" that capture biological phenomena beyond training data (Ustyuzhaninov et al., 2022; Wang et al., 2025). With the shift to transformers, new variants have explored ViT cores (Li et al., 2023), hybrid convolution-attention designs (Lin et al., 2024; Pierzchlewicz et al., 2023), and spatial-transformer readouts (Saha et al., 2024), though most still omit video input.

Transformer models were also applied to response-to-response modeling, beginning with NDT (Ye & Pandarinath, 2021), extended to multiple animals (Ye et al., 2023) and neuronal masking (Zhang et al., 2024). STDNT (Le & Shlizerman, 2022) extended NDT by explicitly modeling correlations across neurons but it did not consistently outperform NDT. While NDT projects all neurons together via linear layers, Quantformer (Calcagno et al., 2024), also a transformer-based forecaster, introduced neuron-specific tokens to handle any number of neurons. POYO (Azabou et al., 2023) introduced

single-neuron tokenization for behavior decoding, removing the need for time-window binning, and POYO+ (Azabou et al., 2025) extended this to discrete classification tasks such as stimulus orientation. POCO (Duan et al., 2025) combined POYO and NDT tokenization to predict neuronal activity from neural history and sub-populations. Representing the most significant scaling of this framework, NEDS (Zhang et al., 2025) modeled approximately 30,000 neurons across 74 sessions with a multi-task loss. None of these models incorporate visual stimuli.

To study the combined effect of brain state and visual input on neuronal responses, Bashiri et al. (2021) used a CNN branch for static stimuli with an additional flow-branch for trial-to-trial correlations. For dynamic video stimuli, Schmidt et al. (2025) modeled a latent brain state probabilistically, using NDT-based response tokenization. Similarly, Neuroformer (Antoniades et al., 2024) used past activity and visual input but is limited to single sessions and cannot flexibly condition on subsets of neurons or response history. CEBRA (Schneider et al., 2023), a contrastive encoder, also mapped activity to behavior or stimuli, accounting for inter-neuron correlations. The closest analog to OmniMouse outside single-cell neuroscience is d'Ascoli et al. (2025), who predicted fMRI responses from concatenated video, text, and audio embeddings.

**General scaling laws in deep learning.** Large-scale models in language and vision exhibit predictable improvements with scale, described by empirical "scaling laws". Kaplan et al. (2020) first showed that performance follows power-law trends in model size, dataset size, and compute. Hoffmann et al. (2022) refined this with "Chinchilla scaling", prescribing proportional growth of model and data size for optimal efficiency. Whether these frameworks extend to scientific domains, where data are multi-modal, complex, noisy, and limited, is less clear. Examples such as AlphaFold3 (Abramson et al., 2024) suggest that systematic scaling of both models and datasets can drive major advances in AI for science.

**Scaling neuroscience models.** There is no consensus on whether classic machine learning scaling laws apply to single-neuron data. Jiang et al. (2025) questioned their applicability, analyzing the NDT-based model of Zhang et al. (2024). Jiang et al. (2025) argued that cross-session variability – and thus implicit data heterogeneity – is crucial for scaling benefits, though it remains unclear if these results generalize to different mouse tasks or model architectures. Again using an NDT-based model but on motor cortex microelectrode data from monkeys and humans, Ye et al. (2025) reported that scaling is constrained by data variability, which pretraining alone cannot fully overcome. Consistent with this view, POCO (Duan et al., 2025) used calcium imaging to show that longer recordings improve predictive performance, aligning with earlier results of Lurz et al. (2021). However, POCO included fewer than 90,000 neurons, mostly from zebrafish ($\sim$77,000). Neural saturation has also been observed: Gokce & Schrimpf (2024) found that behavioral alignment improves with model size, but neural alignment plateaus, with gains concentrated in higher-level visual areas. In contrast, Antonello et al. (2023) reported no such saturation when predicting language and audio fMRI responses, suggesting that scaling limits may depend on the modality and data regime. The largest single-cell response-to-behavior prediction model is POYO+ Azabou et al. (2025) with $\sim$100,000 neurons, which did not perform extensive scaling analyses.

## 3 DATASET

**Neuronal responses.** We compiled a dataset of more than 3 million single-unit neuronal recordings from 73 animals (Fig. 2). The dataset contains excitatory neurons' responses in visual cortex recorded via wide-field two-photon calcium imaging at 6–14 Hz in awake, head-fixed, behaving mice (Sofroniew et al., 2016), with spiking activity extracted by CAIMAN (Giovannucci et al., 2019).

**Visual stimuli.** The mice were presented with naturalistic images sampled from ImageNet (Russakovsky et al., 2015) and videos sampled from cinematic movies and the Sports-1M dataset (Karpathy et al., 2014). In addition, mice were shown parametric stimuli such as static and drifting Gabors (Petkov & Subramanian, 2007), directional pink noise, flashing Gaussian dots, random dot kinematograms (Morrone et al., 2000), and model-generated stimuli (similar to Walker et al., 2019). All stimuli were presented at 30–60 Hz, at varying resolutions with images presented for 500 ms and preceded by a 300–500 ms blank screen.

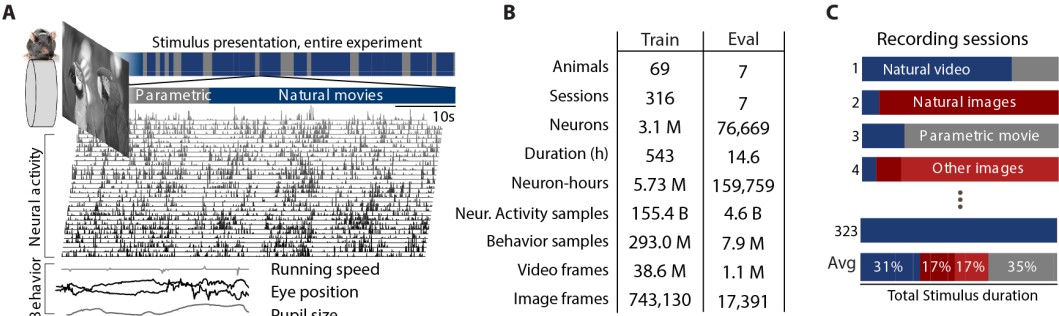

Figure 2: **Data**. **A**. Data were collected via calcium imaging from head-fixed mice running on a wheel while viewing visual stimuli. Behavior variables include pupil center $x$ and $y$ positions, pupil dilation and its derivative and running speed. **B**. Dataset statistics. **C**. Different visual stimuli were presented across sessions, with stimulus types varying by session.

**Behavior variables.** Our dataset contains five behavior variables: running speed, recorded at 50-100 Hz, and four pupil variables: pupil center $x$ and $y$ positions, pupil dilation and its derivative, all recorded at 20 Hz.

**Data utilization.** We reconstruct the visual stimulus presented throughout the entire recording, enabling continuous representation of the full experimental timeline including blank periods across all diverse visual paradigms. For model training, we downsample all behaviors to 20 Hz, visual stimuli to 30 Hz, and linearly upsample all neuronal responses to 30Hz to be comparable to the SENSORIUM 2023 benchmark. All token counts throughout this work refer to the number of samples before upsampling to avoid artificially inflating the dataset size.

## 4 OmniMouse architecture

### 4.1 Tokenization

The input to OmniMouse is a temporal chunk of multi-modal data: neural responses, stimulus frames, and behavioral traces, all time-aligned at their respective sampling rates. This chunk is paired with a *masking configuration* specifying which samples from each modality should be encoded (i.e., unmasked) and which subset of the masked samples should serve as reconstruction targets. After encoding each modality as "token" embeddings, we apply the mask by simply removing masked tokens from the input and constructing query embeddings for the targets. Below we describe this process for each modality.

**Neuronal responses.** To enable per-neuron, per-sample masking, we adopt the tokenization from Azabou et al. (2025); Duan et al. (2025). For $P_i$ neurons (possibly sampled from a larger recorded population) from a session $i$, we construct a tensor of $S_R$ calcium trace samples and apply a strided 1D convolution (stride $s_R$, window $w_R$) along the temporal dimension: $f_{\text{conv}} : \mathbb{R}^{P_i \times S_R} \to \mathbb{R}^{P_i \times T \times D_{\text{model}}}$, where $X_R = f_{\text{conv}}(R)$, $D_{\text{model}}$ is the model's latent dimension, and $T = \lfloor \frac{S - w_R}{s_R} \rfloor + 1$ is the number of tokens per neuron. $s_R$ and $w_R$ control the size and overlap of the token sample-groups.

We augment each token with learned identity embeddings for its neuron, session, and animal, similar to Azabou et al. (2025). To limit the number of per-neuron parameters and decouple it from model size Antolík et al. (2016); Klindt et al. (2017); McIntosh et al. (2016), each embedding table has a fixed dimension $D_{\text{embed}} = 128$ and a corresponding linear projection $W \in \mathbb{R}^{D_{\text{model}} \times D_{\text{embed}}}$ into model space. We flatten the neuron and temporal dimensions and obtain:

$$ID = W_u E_u(N_i) + W_s E_s(i) + W_a E_a(i) \in \mathbb{R}^{P_i \times D_{\text{model}}} \tag{1}$$

$$Z_R = \text{Flatten}\left(X_R + ID\right) \in \mathbb{R}^{P_i T \times D_{\text{model}}} \tag{2}$$

where $N_i \in \mathbb{R}^{P_i}$ contains neuron IDs. The masking configuration for neural activity specifies which tokens to mask and which to reconstruct. These masked and target tokens are removed from $Z_R$

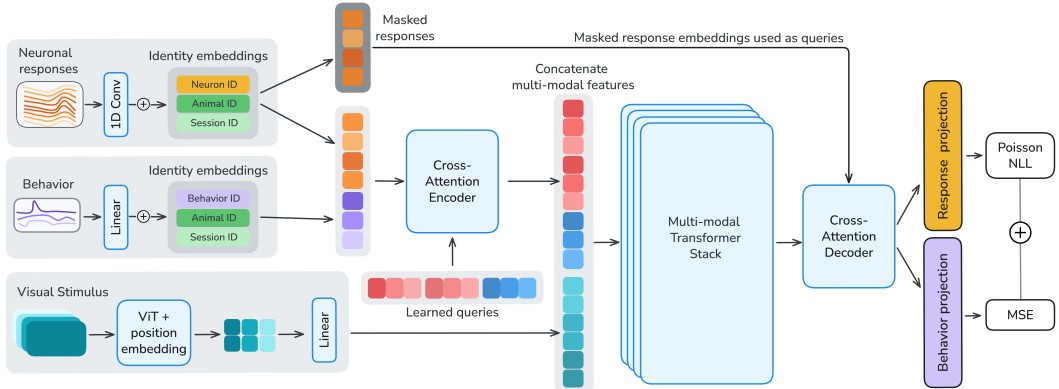

Figure 3: **Model architecture.** OmniMouse introduces a unified framework that handles arbitrary combinations of neural forecasting, sub-population prediction, stimulus encoding, and behavioral decoding through flexible masking. We adopt single-neuron, single-time-chunk tokenization and a cross-attention encoder (following POYO+ (Azabou et al., 2025)), along with analogous queries to the multi-modal cross-attention decoder. A step-by-step summary of the training procedure is provided in Appendix A.4.

to create the encoder input $\widetilde{Z}_R \in \mathbb{R}^{(P_i T - n_{masked}) \times D_{\text{model}}}$. For target tokens, we construct decoder queries by simply using their identity embeddings, yielding $Q_R = ID_{\text{targets}} \in \mathbb{R}^{n_{\text{targets}} \times D_{\text{model}}}$. Each query reconstructs the $s_R$ samples uniquely captured by the corresponding token. In addition to its embedding vector, each token or query is assigned a timestamp indicating its relative temporal position within the chunk context. For neuronal response embeddings, we use the timestamp from the first of its $s_R$ samples.

**Visual stimuli.** Given $S_V$ frames ($\mathbb{R}^{S_V \times H \times W}$) of grayscale video, we use the first ten layers of a randomly-initialized Hiera vision transformer Ryali et al. (2023) followed by a linear projection to extract spatiotemporal embeddings $Z_V \in \mathbb{R}^{H'W'S'_V \times D_{\text{model}}}$, where $H'$, $W'$, and $S'_V$ result from Hiera's convolutional patch embedding and hierarchical pooling.

The masking configuration specifies a contiguous unmasked frame range to encode. Because Hiera's early stages use local temporal attention, each output token has a well-defined temporal window, yielding the final encoder input, $\widetilde{Z}_V$. Each video embedding is assigned the timestamp from the first frame within its temporal receptive field.

**Behavioral traces.** For $S_B$ samples from each of the five behavioral variables, we construct a single behavior tensor ($\mathbb{R}^{5 \times S_B}$), which is either completely masked or completely unmasked. When unmasked, we use a shared linear layer to project the traces along the temporal dimension and add learned channel-specific embeddings (as well as the appropriate session and animal ID embeddings), yielding encoder input sequence $Z_B \in \mathbb{R}^{5 \times D_{\text{model}}}$. When behavior is masked, we instead use it as a target and construct queries with the channel and session/animal ID embeddings $Q_B \in \mathbb{R}^{5 \times D_{\text{model}}}$, which are passed to the shared decoder to reconstruct each channel's full trace. Because each behavioral token/query captures the entire context window, we simply assign them to timestamp 0.

## 4.2 MASKING

We train OmniMouse with 119 structured masking configurations (Tab. S6), including our core evaluation tasks (forecasting, population prediction, stimulus encoding, and behavioral decoding) as well as numerous systematic variations that reduce or combine context across modalities.

**Neural responses.** We define a consistent prediction target used by all masking configurations—the final 1 second (30 samples) of activity from 3072 randomly selected neurons—and vary unmasked context along two axes: *Population context* provides contemporaneous activity from a subset of non-target neurons, while *causal context* provides temporally preceding activity from all neurons. We

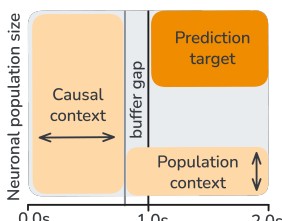

Figure 4: Masking of neuronal responses

also combine masking along these axes (i.e., provide a causal context from only a subset of neurons), forcing the model to interpolate along both the population and temporal dimensions. To maintain consistent shapes across sessions, we first randomly sample 4096 neurons (of which 3072 become targets) per session before applying masks. Finally, we enforce a 5-sample buffer gap Fig. 4 between causal context and prediction target to prevent any upsampling artefacts.

**Visual stimuli.** As noted above, we define a visible frame range, and vary the start time and duration. Frames preceding the target enable forecasting from video, while contemporaneous frames enable stimulus encoding tasks.

**Behavior.** We either fully provide behavioral traces as encoder input or fully mask them and use them as decoder reconstruction targets.

## 4.3 MODEL

After tokenization, we process the multi-modal embeddings through three stages: (1) an encoder that compresses unmasked neuronal and behavioral tokens into a fixed-length latent sequence, (2) a multi-modal fusion stack that integrates these latents with features of the visual stimulus, and (3) a decoder that reconstructs target neuronal responses and behavioral traces from the fused representation. Full hyperparameter specifications are provided in App. A.1.

**Encoding.** First, we concatenate the unmasked neuronal response and behavioral embeddings into a single input sequence, $Z_{RB} = \left[ \tilde{Z}_R, Z_B \right]$. Note that depending on the masking configuration, the length of $\tilde{Z}_R$ will vary and $Z_B$ may be absent (when behavior is masked). Following Azabou et al. (2023; 2025), we compress this variable-length sequence into a fixed set of latent embeddings via cross-attention with learned queries. We use $M$ unique query embeddings, repeating each $N$ times, for $N$ different timestamps uniformly spaced across the context window, yielding $M \times N$ query embeddings total. To each of the unique embeddings, we also add the session and animal identity embeddings. We use a slightly larger $M$ than in prior work to avoid an information bottleneck with larger neural populations.

Within the cross-attention block, we implement *local sliding-window attention*: each query and input token is assigned a local temporal window (with length determined by its modality), and attention is masked between any pair of features whose windows do not overlap. To retain global information flow and avoid attention sinks, we additionally append $G$ *global register* tokens (Darcet et al., 2023) to the query embeddings, which attend to the entire key sequence. Depending on the masking configuration, we remove query repeats for portions of the context window with no overlapping input features.

**Multi-modal fusion.** We concatenate the encoded latent sequence with any unmasked video features from the Hiera encoder and pass the result through a stack of $L$ multi-modal transformer layers (Tab. S5), consisting of self-attention and a feed-forward network. We interleave layers with *local sliding-window attention* and global layers with no attention masking at a ratio of 5:1. This design allows efficient processing of long sequences while periodically enabling full cross-modal and long-range temporal interaction. All transformer layers throughout the model (including the encoding and decoding cross attention blocks) use 1D-RoPE (Su et al., 2024) computed from each token's timestamp, to encode relative timing both within and across modalities.

**Decoding.** We extend the cross-attention decoder of Azabou et al. (2025) to support neuronal response prediction alongside behavioral decoding. The fused multi-modal features from the final transformer layer serve as keys and values in a shared decoder cross-attention block. As queries, we concatenate the neuronal target queries $Q_R$ and, when behavior is a prediction target, the behavioral queries $Q_B$. No explicit task or modality embeddings are added. This cross-attention also uses local causal sliding-window attention, where each query attends only to latent features within its temporal neighborhood. After cross-attention and a shared feed-forward layer, the outputs are routed to modality-specific linear readouts that project from $d_M$ back to the target dimensionality: $s_R$ samples per neuronal query and $S_B$ behavioral samples per behavioral query.

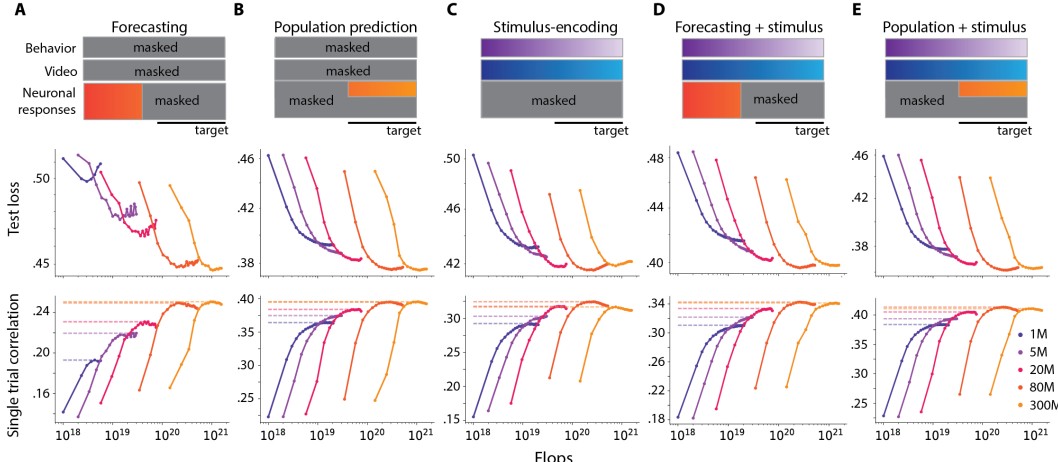

Figure 5: **Task-specific performance gains with model scaling.** Top row: masking schema. Middle row: Test loss. Bottom row: single-trial correlation. **A**. Forecasting; predicting one second neuronal activity, conditioned only on past neuronal activity **B**. Population prediction; conditioned on a sub-population of $N = 1024$ neurons. **C**. Stimulus-encoding: Neuronal encoding conditioned on the visual stimulus. **D**. Stimulus-conditioned forecasting: same as forecasting, but also conditioned on the visual stimulus. **E**. Stimulus-conditioned population prediction; same as population prediction ($N = 1024$ neurons in population context), with additionally provided context of visual stimulus.

## 4.4 TRAINING

We trained our model to predict both neuronal responses and behavioral traces, using Poisson loss (averaged across neurons) for neural encoding and mean squared error (MSE) loss for behavior decoding. To balance the two objectives, the behavioral loss is down-weighted by a factor of 0.1 so that its scale matches the magnitude of the Poisson loss. For our scaling experiments, we trained models end-to-end on either the complete dataset of 323 sessions or on constructed collections of 8, 16, 32 or 64 sessions to study data scaling effects. These nested collections were designed so that larger collections always contained all sessions from smaller ones, ensuring consistent evaluation .

We followed Hu et al. (2024); Wen et al. (2024); Hägele et al. (2024) and trained our model with warmup followed by a constant learning rate for at least 250k steps, saving checkpoints every 20k steps. This warmup-stable training phase serves a dual purpose: it trains the model to convergence while simultaneously producing a dense series of intermediate checkpoints that span a wide range of compute budgets. To obtain a clean final evaluation point from each checkpoint, we continue training from each saved checkpoint for an additional 10k steps using an inverse-square-root learning rate decay, which anneals the learning rate to near-zero and allows the model to settle into a local optimum. Each such decayed checkpoint then yields one evaluation point in our scaling plots. This procedure allows us to densely populate the compute axis of our scaling curves without training separate models from scratch at each compute budget. See App. A.5 for infrastructure details.

## 5 UNIFIED EVALUATION FRAMEWORK

All scaling experiments use a standardized evaluation protocol on the same mice to ensure fair comparison across models, baselines, and conditions. We chose seven mice (*evaluation mice*) comprised of five publicly available datasets from SENSORIUM 2023 and two test mice from SENSORIUM 2022. For all analyses, we use the held-out set provided by these datasets. As in model training, we define a consistent prediction target used by all evaluation configurations: the final second (30 samples) of neuronal activity of randomly chosen 3072 neurons in each session are the prediction target. We evaluate five response prediction tasks (Fig. 5) as well as a behavior decoding task (Fig. 6):

**Forecasting**. Predicting one second of neuronal activity given the previous second (i.e. one second of causal context only). We use MtM (Zhang et al., 2024), a model that supports causal and population masking, as the baseline.
**Population prediction**. Provided with a population context of unmasked $N = 256$ neurons over the full 2-second context window, predict a target population. This task assesses how much of the trial-to-trial variability can be explained by simultaneously recorded neurons. As in the forecasting

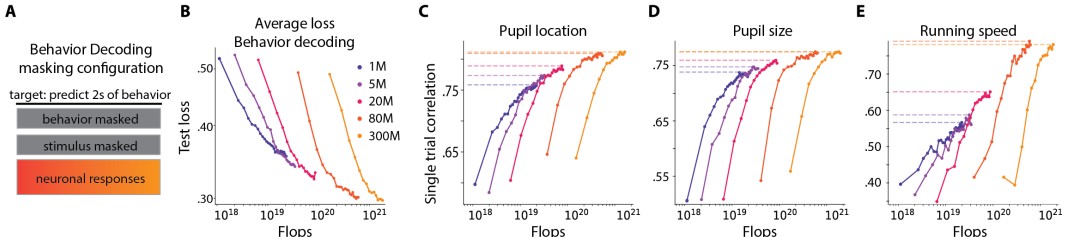

Figure 6: **Behavior decoding scales with model size. A**. Masking for behavior decoding. **B**. Decoding loss averaged across all behaviors variables. **C-E**. Prediction performance for behavioral decoding.

task, we chose MtM (Zhang et al., 2024) as the baseline model. For the scaling analysis in Fig. 5, we provide $N = 1024$ neurons as population context.

**Stimulus-encoding**. Given the full two seconds of visual stimulus, predict neuronal activity for the final second. The neuronal encoding competitions SENSORIUM 2023 (Turishcheva et al., 2024) and SENSORIUM 2022 (Willeke et al., 2022) establish strong baselines for this task.

**Stimulus-conditioned forecasting**. This task additionally provides the full two seconds of visual stimulus as context to the forecasting task. We use Schmidt et al. (2025) as a baseline model, which also conditions response prediction on neurons and video.

**Stimulus-conditioned population prediction**. With the full two seconds of visual stimulus as context in addition to $N = 256$ unmaksed neurons. Again, we use Schmidt et al. (2025) as the baseline.

**Behavior decoding**. Given the activity of all neurons for the full two seconds, predict all behavioral traces. CEBRA (Schneider et al., 2023) and Poyo+ (Azabou et al., 2025) are used as baselines.

For a fair comparison, we train the baselines on the same amount of data as OmniMouse. We use the smallest collection of eight mice from data-scaling experiment (Fig. 7) as a data-matched comparison. Implementation details and hyperparameters for each baseline are provided in App. B. For all experiments, consistent with SENSORIUM 2022/2023 competitions, we opt for *single-trial correlation*, i.e. the pearson correlation between ground truth and predicted signal, as the single evaluation metric, without any correction for the noise ceiling. This measure allows for an unbiased comparison of predictivity across all tasks and conditions.

## 6 RESULTS: THE BENEFITS OF SCALING

**Current neuronal-predictive models are not compute- or parameter-limited.** Because collecting neuronal data is costly, we first asked if existing models are already limited by compute or parameters, or if more data would still improve performance. To answer this question, we trained models on all 323 sessions while scaling width and depth as in Tab. S5. We evaluated five neuronal response masking strategies (Fig. 5, top row): two based on response dynamics (forecasting and population con-

Table 1: **Baseline comparisons.** Results displayed in **bold** indicate the highest score per task in either the data-matched condition (top) or when using the full dataset (bottom). Baselines were evaluated for all conditions that they support, with ✗ denoting an unsupported condition. Conditions: Forecasting (*Fcst*), forecasting + stimulus (*Fcst+S*), population prediction (*Pop*) with $n = 256$ visible neurons, population prediction + stimulus (*Pop+S*). Behavioral decoding: Average across behaviors (*Avg*), pupil-location (*Gaze*), pupil-size (*Pupil*), and running speed (*Running*).

|  | Model | Neuronal Activity Prediction | | | | Behavior Decoding | | | |
|---|---|---|---|---|---|---|---|---|---|
|  |  | Fcst | Fcst+S | Pop | Pop+S | Avg | Gaze | Pupil | Running |
| **8 sessions** | **MtM** (Zhang et al., 2024) | 0.12 | ✗ | 0.07 | ✗ | ✗ | ✗ | ✗ | ✗ |
|  | **Latent Model** (Schmidt et al., 2025) | ✗ | 0.18 | ✗ | 0.16 | ✗ | ✗ | ✗ | ✗ |
|  | **CEBRA** (Schneider et al., 2023) | ✗ | ✗ | ✗ | ✗ | 0.53 | 0.52 | 0.55 | **0.51** |
|  | **POYO+** (Azabou et al., 2025) | ✗ | ✗ | ✗ | ✗ | 0.55 | 0.56 | 0.63 | 0.47 |
|  | **OmniMouse-5M** | **0.18** | **0.25** | **0.25** | **0.27** | **0.59** | **0.68** | **0.66** | 0.44 |
| **323 sessions** | **OmniMouse-1M** | 0.18 | 0.31 | 0.27 | 0.35 | 0.68 | 0.75 | 0.73 | 0.55 |
|  | **OmniMouse-5M** | 0.22 | 0.32 | 0.28 | 0.35 | 0.69 | 0.76 | 0.74 | 0.57 |
|  | **OmniMouse-20M** | 0.23 | 0.33 | 0.29 | **0.37** | 0.75 | 0.78 | 0.75 | 0.73 |
|  | **OmniMouse-80M** | **0.25** | **0.34** | 0.29 | **0.37** | **0.77** | **0.80** | **0.76** | **0.75** |
|  | **OmniMouse-300M** | **0.25** | **0.34** | **0.30** | **0.37** | 0.76 | **0.80** | **0.76** | 0.73 |

Table 2: **Sensorium benchmark results.** Comparison against Sensorium 2022 and 2023 competition winners across both tracks. See table S7 for additional comparisons.

| | Sensorium 2022 | | Sensorium 2023 | |
| --- | --- | --- | --- | --- |
| | Main | Bonus | Main | Bonus |
| Competition Winner | 0.33 | **0.45** | 0.29 | 0.22 |
| OmniMouse-80M | **0.37** | **0.45** | **0.33** | **0.30** |

text), two analogous variants that additionally condition on video (video-conditioned forecasting and video-conditioned population context), and one stimulus-encoding strategy (video & behavior). For each strategy, models ranged from 1M to 300M parameters, and we tracked both test loss and single-trial correlation as a function of total compute (model FLOPs, excluding FLOPs of neuron-specific parameters). Performance improved across all neuronal prediction tasks as model size increased up to 80M parameters (Fig. 5). Beyond this point, gains were minimal, as loss curves saturated or overfit, indicating that current models are data-limited rather than compute- or parameter-limited.

**OmniMouse achieves state-of-the-art performance.** Our large-scale model outperforms all baselines across six evaluation regimes for both response prediction and behavior decoding (Tab. 1). Crucially, these gains are not simply due to training on more data: in data-matched comparisons, where OmniMouse and baselines are trained and evaluated on identical datasets, our model outperforms strong specialized methods across all tasks except for decoding of running speed. This demonstrates that the architectural and masking design of OmniMouse provides advantages independent of data scale. OmniMouse also sets a new state of the art on the Sensorium 2022 and 2023 competitions (Tab. 2). For the Sensorium 2023 competition, we evaluate our model (1) with a frozen pretrained OmniMouse-80M backbone and only neuron-specific parameters trained, and (2) with full end-to-end training on the same 10-mouse competition dataset (see Tab. S7 for details), and surpass the prior state of the art in both cases. We note that while our data-matched setting uses the same training data, our framework additionally enables training across video boundaries — an information not available for the previous models.

**Behavior prediction shows the most promising scaling dynamics on the available data.** To characterize the scaling of behavior decoding, we used the same models and evaluated their ability to predict pupil location, pupil size, and running speed from neuronal activity only (Fig. 6). Across all three settings, performance improved smoothly with compute budget, reminiscent of classic scaling-law behavior. Larger models consistently achieved higher single-trial correlations, albeit with an indication of saturation at the largest scale tested. The models had not yet fully converged for the behavioral decoding task, and longer training could have improved performance further, even on the largest model. These results show that behavioral prediction continues to improve with model scaling and may benefit from further increases in capacity.

**Scaling dataset size improves performance.** To study how dataset size affects performance, we trained three model sizes – 5M, 20M, and 80M – on nested collections of 8, 16, 32, 64, and 323 sessions, such that the larger collections are supersets of the smaller ones (Fig. 7A). For evaluation, we test the model on the same held-out test set of the same seven mice that were contained in all

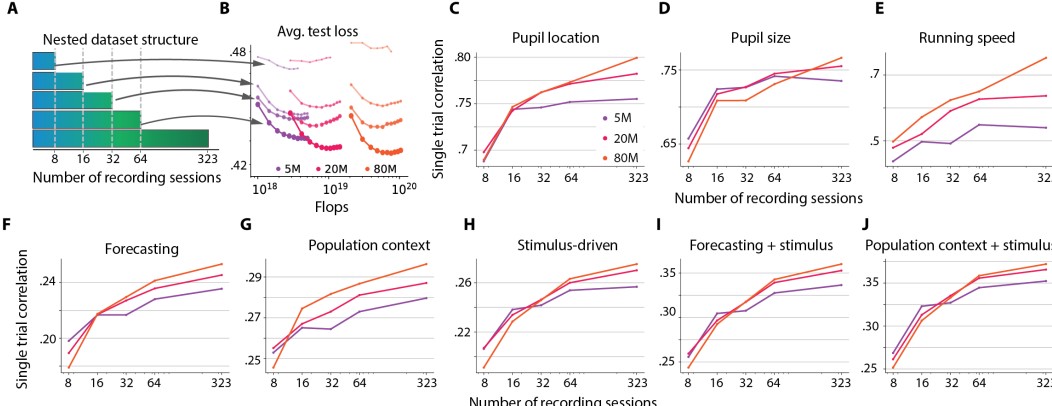

Figure 7: **Scaling data improves model performance. A**. Nested datasets structure. **B**. Test loss for different model and data sizes, averaged across all response prediction tasks. **C-K**. Performance improvements for all tasks when scaling dataset from 8 to 323 sessions.

collections (Fig. 7C–J). In all cases, performance improved with the number of sessions, exhibiting predictable data-scaling trends. Larger models consistently benefited more from additional data. The larger models required a minimum size of the training set to outperform the smaller models and the performance gap widened as the dataset increased in size. Behavior decoding benefited the most from data scaling (Fig. 7C–E), showing no saturation and large performance differences between 5M and 80M models. For responses, the strongest gains were observed for tasks that included video input (Fig. 7C–E), where the 80M models continued to improve even beyond 100 sessions, suggesting that they remained data-limited rather than capacity-limited. The *forecasting* and *population context* tasks showed larger benefits from scaling of both data and model sizes. The gaps between 20M and 80M models (Fig. 7A, B) increased faster compared to the tasks with video input, which could indicate a lack of diversity of the visual stimuli in our dataset. Overall, these results highlight that scaling both model size and data quantity is synergistic and necessary to approach peak predictive performance.

**OmniMouse enables systematic evaluation of how neuronal context shapes predictive performance.** Lastly, we assessed the model's generalization by testing on masking conditions not seen during training (Fig. S1), varying causal and population context size. Performance scaled smoothly with additional context, demonstrating that OmniMouse learns generalizable representations that enables systematic analyses of contextual contributions to neural variability (see Fig. S2, Fig. S3, and App. C). The ability to flexibly control and combine different sources of context within a single model opens an exciting avenue for future study, enabling precise quantification of how each factor shapes neural variability and their interactions across brain areas, stimuli, and behavioral states.

## 7 DISCUSSION

In this work, we introduce OmniMouse, a multi-modal, multi-task model of mouse visual cortex that jointly integrates neural activity, video stimuli, and behavioral traces across animals within a single unified architecture. Trained on one of the largest single-neuron dataset to date, OmniMouse achieves state-of-the-art performance across tasks, outperforming strong specialized baselines on neural response prediction, activity forecasting, and behavioral decoding. The breadth and scale of both model and dataset enable a systematic study of scaling behavior in single-neuron brain models.

Our motivation for studying scaling laws is practical: if brain models are to become foundation models for neuroscience, it is essential to ask whether current data can sustain scaling. Despite using naturalistic movies and images, we find that performance saturates with model size, suggesting data – not compute – as the limiting factor. This is consistent with recent observations that scaling benefits in neural models are constrained by data variability rather than model capacity (Jiang et al., 2025; Ye et al., 2025; Gokce & Schrimpf, 2024). Even in the relatively simple mouse visual system, richer tasks, more varied stimuli, and larger-scale recordings are needed to support continued scaling. At the same time, relatively sparse sampling already yields strong models: with 60,000 neurons from eight mice, predictive accuracy is high, likely due to redundancy in neural codes (Stringer et al., 2019). Additional gains from larger datasets appear modest, paralleling language (Kaplan et al., 2020; Hoffmann et al., 2022) – yet in those domains, small improvements have triggered phase transitions to qualitatively new abilities (Wei et al., 2022). By analogy, richer neuroscience data may similarly unlock new capabilities in brain models, revealing deeper principles of neural computation.

**Limitations.** Our work has several limitations. First, OmniMouse parameters scale linearly with the number of neurons, as it learns per-neuron embeddings. This makes training computationally expensive and may limit scaling to larger datasets. Second, large-scale transformers remain difficult to interpret and prone to overparameterization issues, which constrain the biological insights that can be drawn. Lastly, the behavioral data present in our data is limited to spontaneous activity and it is thus unclear if this approach can transfer to more complex behaviors.

**Future work.** Future work could extend to stimulus decoding (Benchetrit et al., 2023; Bauer et al., 2024; Zhu et al., 2025) and more precise study of training dynamics of modality interactions and multi-task learning to improve the masking recipe. Beyond calcium imaging in mouse visual cortex, models could integrate other data types such as electrophysiological recordings, diverse animal species, and more multi-modal stimuli such as audio. Finally, jointly modeling visual input, neuronal responses, and behavior enables analysis of spontaneous and evoked activity (Stringer et al., 2019), revealing how brain state shapes sensory processing and core principles of computation.

REPRODUCIBILITY STATEMENT

To ensure the reproducibility of our results, we provide the complete source code for our multi-modal model, including scripts for training, evaluation, fine-tuning, and inference, available at `https://github.com/enigma-brain/omnimouse` upon publication. Regarding the dataset, which consists of large-scale neuronal responses from the visual cortex and naturalistic visual stimulation, we have detailed the data acquisition and processing pipeline in Appendix E.1. While the full dataset is currently undergoing final preparation due to its unprecedented scale, we are committed to releasing it publicly within six months.

ACKNOWLEDGEMENTS

This work was generously supported by The James Fickel Enigma Project Fund. AST acknowledges the support of the National Science Foundation and of DoD OUSD (R&E) under Cooperative Agreement DBI-2229929 (The NSF AI Institute for Artificial and Natural Intelligence). This work was also supported by the Intelligence Advanced Research Projects Activity (IARPA) via Department of Interior/Interior Business Center (DoI/IBC) contract number D16PC00003. This work was also supported by the German Research Foundation (SFB 1233, Robust Vision: Inference Principles and Neural Mechanisms, project number 276693517; SFB 1528 Cognition of Interaction, project number 454648639; project IDs 432680300 (SFB 1456, project B05) and 515774656, the European Research Council (ERC) under the European Union's Horizon Europe research and innovation programme (grant agreement numbers 101041669 and 101171526). FHS is further supported by the German Federal Ministry of Education and Research (BMBF) via the Collaborative Research in Computational Neuroscience (CRCNS) (FKZ 01GQ2107). FHS and ASE acknowledge the support of the Ministry of Science and Culture of Lower Saxony through funds from the program zukunft.niedersachsen of the Volkswagen Foundation for the "CAIMed – Lower Saxony Center for Artificial Intelligence and Causal Methods in Medicine" project (grant no. ZN4257). This research used resources of the Oak Ridge Leadership Computing Facility at the Oak Ridge National Laboratory, which is supported by the Office of Science of the U.S. Department of Energy under Contract No. DE-AC05-00OR22725. KFW and AG sincerely thank Craig Kapfer, Kurt Stine, and the whole Marlowe team at Stanford, for the support and early access to the supercomputing cluster. We are also thankful for the support of Zoe Ryan and Bruce McGowan and the NVIDIA support team for guidance during all stages of this project. PT, MAW and MV thank the computing time granted by the Resource Allocation Board and provided on the supercomputer Emmy/Grete at NHR-Nord@Göttingen as part of the NHR infrastructure. Additional compute for this work was conducted with computing resources under the projects nim00010 and nim00012. We also thank Surya Ganguli, Dan Yamins, and the entire team of the Enigma Project at Stanford for helpful discussions.

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

# A  IMPLEMENTATION DETAILS

We provide a complete specification of the hyperparameters, masking configurations, and training procedure used for OmniMouse.

## A.1  HYPERPARAMETERS

Tables S1–S4 summarize the full set of hyperparameters for the final OmniMouse model. We organize them into four groups: data and tokenization (Tab. S1), which specifies how each modality is sampled, chunked, and converted into token embeddings; encoder latent queries (Tab. S2), which defines the learned bottleneck between the variable-length input and the fixed-length neuronal and behavioral latent sequence; transformer architecture (Tab. S3), which covers the shared configuration of the encoder, fusion, and decoder stages; and training (Tab. S4), which lists the loss functions, optimizer settings, and learning rate schedule.

Table S1: **Data & tokenization hyperparameters.**

| Parameter | Description | Value |
|---|---|---|
| ***Neural responses*** | | |
| Sampling rate | | 30 Hz |
| $S_R$ | Samples per context window | 60 |
| $w_R$ | Conv. kernel size (samples) | 10 |
| $s_R$ | Conv. stride (samples) | 5 |
| local-attention window size | Attention mask local sliding-window size | .333 s |
| $P_i$ | Neurons sampled per batch | 4096 |
| Interpolation buffer | Gap between context and targets (samples) | 5 |
| Num. reconstructed neurons | Size of masked sub-population to be reconstructed | 3072 |
| Num. reconstructed samples | Duration of reconstruction target (samples) | 30 |
| ***Visual stimuli*** | | |
| Frame rate | Video frame rate | 30 fps |
| $S_V$ | Frames per context window | 60 |
| local-attention window size | Attention mask local sliding-window size | .2 s |
| $C \times H \times V$ | Spatial resolution | $1 \times 36 \times 64$ |
| *Hiera encoder* | | |
| Layers per stage | Hierarchical stage depths | (2, 8) |
| Embedding dimension | Output dimension of patch embedding | 128 |
| Num. attention heads | | 2 |
| MLP ratio | FFN hidden dim / embed dim | 2.67 |
| Patch kernel $(t, h, w)$ | Convolutional patch embedding | (6, 7, 7) |
| Patch stride $(t, h, w)$ | Patch embedding stride | (2, 2, 2) |
| Patch padding $(t, h, w)$ | Patch embedding padding | (2, 1, 3) |
| Query stride $(t, h, w)$ | Pooling stride per stage | (1, 2, 2) |
| Mask unit size $(t, h, w)$ | Local attention window | (1, 8, 8) |
| $(T' \times H' \times W' \times D')$ | Output feature shape (pre-projection) | (30, 8, 16, 768) |
| Drop path rate | | 0.25 |
| ***Behavioral traces*** | | |
| Sampling rate | Behavior sampling rate | 20 Hz |
| $S_B$ | Samples per context window | 40 |
| Channels | Eye tracker (4) + treadmill (1) | 5 |
| local-attention window size | Attention mask local sliding-window size | 2 s |

Table S2: **Encoder latent query hyperparameters.**

| Parameter | Description | Value |
|---|---|---|
| $M$ | Unique latent query groups | 640 |
| $N$ | Repeats per group (covers context window) | 6 |
| $G$ | Global register tokens | 256 |
| Total latents | $M \times N + G$ | 4096 |
| local-attention window size | Attention mask local sliding-window size (excluding globals) | .333 s |

Table S3: **Transformer architecture hyperparameters.** Values of hyperaparameters that vary with scaling are indicated with ? (see table Tab. S5).

| Parameter | Description | Value |
|---|---|---|
| $d_e$ | Identity embedding dimension | 256 |
| $d_m$ | Model / latent dimension | ? |
| $L$ | Multi-modal transformer layers | ? |
| $h$ | Number of attention heads (all stages) | ? |
| $d_h$ | Attention head dimension | $d_m/h$ |
| Normalization type | | RMSNorm |
| Norm $\epsilon$ | | $10^{-5}$ |
| Pre-norm | Normalization applied before attention/FFN | ✓ |
| QK-normalization | Normalization applied to Q and K projections | ✓ |
| Drop path rate | Stochastic depth | 0.0 |
| FFN hidden multiplier | FFN hidden dim / $d_m$ | 2.67 |
| FFN activation | Gated activation function | SiLU (SwiGLU) |
| Local:global ratio | Interleaving ratio of *local sliding-window attention* layers | 5:1 |
| Response output activation | Activation on neural predictions | Softplus($\beta$=0.1) |
| Behavior output activation | Activation on behavior predictions | None (identity) |

Table S4: **Training hyperparameters.**

| Parameter | Description | Value |
|---|---|---|
| *Loss* | | |
| Neural loss type | | Poisson negative log-likelihood |
| Behavior loss | | Mean squared error |
| $\lambda_B$ | Behavior loss down-weighting factor | 0.1 |
| *Optimization* | | |
| Optimizer | | AdamW |
| Learning rate | | $7 \times 10^{-4}$ |
| Warmup steps | Linear warmup duration | 20 000 |
| Learning rate decay type | | Inverse square root (10k steps) |
| Learning rate decay steps | | 10k steps |
| $\beta_1, \beta_2$ | Adam momentum parameters | 0.9, 0.98 |
| Weight decay | L2 regularization | 0.1 |
| Gradient clipping type | | Norm |
| Gradient clipping value | | 1.0 |

## A.2 MODEL SCALING

Tab. S5 summarizes the five model scales used in our scaling experiments, ranging from $\sim$ 1M to $\sim$ 300M parameters. We scale along two primary axes: model width ($d_m$) and depth ($L$, the number of transformer layers in the multi-modal fusion stack). Importantly, because we decouple the identity embedding dimension $d_e$ from the model width via learned linear projections (as described in Sec. 4), the number of neuron-, session-, and animal-specific parameters ($p_N$) remains fixed across all scales.

Table S5: **Scaling variants of OmniMouse.** $L$: multi-modal transformer layers; $d_m$: model dimension; $h$: number of attention heads; $d_e$: dimensions of all embeddings; $p_L$: multi-modal transformer layer parameters; $p_M$: model parameters (excluding neuronal embeddings); $p_N$: all neuronal, session, and animal parameters; $p_T$: total parameters; $S$: sequence length.

| Model | $L$ | $d_m$ | $h$ | $d_e$ | $p_L$ | $p_M$ | $p_N$ | $p_T$ | $S$ |
|---|---|---|---|---|---|---|---|---|---|
| OmniMouse-1M | 2 | 256 | 4 | 256 | 1.7M | 6M | 779M | 885M | 4096 |
| OmniMouse-5M | 6 | 256 | 8 | 256 | 5.1M | 10.4M | 779M | 891M | 4096 |
| OmniMouse-20M | 6 | 512 | 8 | 256 | 19.1M | 29.1M | 779M | 810M | 4096 |
| OmniMouse-80M | 12 | 768 | 12 | 256 | 88M | 115M | 779M | 894M | 4096 |
| OmniMouse-300M | 24 | 1024 | 16 | 256 | 308M | 348M | 779M | 1.1B | 4096 |

## A.3 MASKING CONFIGURATIONS

Tab. S6 enumerates all 119 structured masking configurations used during training. All configurations share a fixed prediction target: the final 30 samples (1 s) of activity from up to 3072 randomly selected neurons. Rows enumerate the systematic variations over population context, causal context, video context, and behavior encoding/decoding. Entries in brackets denote multiple configurations swept over the listed values.

Table S6: **Masking configurations.**

| Mask | Behavior | Video (last frames visible) | Visible Neurons | Pop. Context (from → to) | Causal Context (from → to) | Predicted Behavior |
|---|---|---|---|---|---|---|
| 1–3 | ✓ | 0 | [64, 256, 1024] | 0 → 60 | — | |
| 4 | ✗ | 0 | 4096 | 0 → 60 | — | ✓ |
| 5–7 | ✗ | 0 | [64, 256, 1024] | 0 → 60 | — | ✓ |
| 8–19 | ✓ | 0 | [64, 256, 1024, 4096] | — | [0, 10, 15] → 25 | |
| 20–28 | ✓ | 0 | [64, 256, 1024] | 25 → 60 | [0, 10, 15] → 25 | |
| 29–37 | ✗ | 0 | [64, 256, 1024] | 25 → 60 | [0, 10, 15] → 25 | ✓ |
| 38–40 | ✓ | 10 | [64, 256, 1024] | 10 → 60 | — | |
| 41–52 | ✓ | 10 | [64, 256, 1024, 4096] | — | [0, 10, 15] → 25 | |
| 53–58 | ✓ | 10 | [64, 256, 1024, 4096] | 25 → 60 | [10, 15] → 25 | |
| 59–61 | ✓ | 20 | [64, 256, 1024] | 20 → 60 | — | |
| 62–73 | ✓ | 20 | [64, 256, 1024, 4096] | — | [0, 10, 15] → 25 | |
| 74–79 | ✓ | 20 | [64, 256, 1024] | 25 → 50 | [10, 15] → 25 | |
| 80–82 | ✓ | 20 | [64, 256, 1024] | 30 → 60 | — | |
| 83–94 | ✓ | 30 | [64, 256, 1024, 4096] | — | [0, 10, 15] → 25 | |
| 95–100 | ✓ | 30 | [64, 256, 1024] | 25 → 40 | [10, 15] → 25 | |
| 101–103 | ✓ | 40 | [64, 256, 1024] | 30 → 50 | — | |
| 104–111 | ✓ | 40 | [64, 256, 1024, 4096] | — | [10, 15] → 25 | |
| 112–114 | ✓ | 50 | [64, 256, 1024] | 30 → 40 | — | |
| 115–118 | ✓ | 50 | [64, 256, 1024, 4096] | — | 10 → 20 | |
| 119 | ✓ | 60 | — | — | — | |

## A.4 OMNIMOUSE TRAINING ALGORITHM

Algorithm App. A.4 provides a concise summary of the full OmniMouse training loop, from tokenization through loss computation. At each training step, a single masking configuration is sampled uniformly along with a batch of multi-modal data *chunks* from a single session.

---

**Algorithm 1** Train OmniMouse

---

**Input:** neural responses $R$, video frames $V$, behavioral traces $B$, masking set $\mathcal{M}$, loss weight $\alpha$
Sample masking configuration $m \sim \mathcal{M}$ uniformly
*// Tokenization*
Tokenize $R$ via strided 1D convolution; add neuron, session, and animal identity embeddings $\rightarrow Z_R$
Tokenize $V$ via Hiera encoder and linear projection $\rightarrow Z_V$
Tokenize $B$ via linear projection; add channel, session, and animal identity embeddings $\rightarrow Z_B$
*// Masking*
Remove masked tokens from $Z_R \rightarrow$ encoder input $\widetilde{Z}_R$
build decoder queries from target identity embeddings $\rightarrow Q_R$
Remove out-of-range tokens from $Z_V \rightarrow$ encoder input $\widetilde{Z}_V$
**if** behavior unmasked **then** include $Z_B$ in encoder input
**else** build decoder queries from channel/session/animal embeddings $\rightarrow Q_B$
*// Encoding*
Compress $[\widetilde{Z}_R, Z_B]$ into $M \times N$ latents via cross-attention with learned queries and $G$ global registers
*// Multi-modal Fusion*
Concatenate latents with $\widetilde{Z}_V$
pass through $L$ transformer layers $\rightarrow \mathcal{F}$
*// Decoding*
Decode $[Q_R, Q_B]$ via local causal cross-attention over $\mathcal{F}$
apply modality-specific linear readouts $\rightarrow \hat{R}_{\text{targets}}, \hat{B}$
*// Loss and Update*
Compute $\mathcal{L}(\theta) = \text{PoissonLoss}(\hat{R}_{\text{targets}}, R_{\text{targets}}) + \alpha \cdot \text{MSE}(\hat{B}, B)$
Update $f_\theta$ via gradient descent on $\mathcal{L}(\theta)$

---

## A.5 COMPUTE INFRASTRUCTURE AND TRAINING PROCEDURES

All models were trained on Marlowe (Kapfer et al., 2025), Stanford's NVIDIA DGX H100 SuperPOD cluster. We scaled training from a single node (8 GPUs per node) for smaller models up to 10 nodes for OmniMouse-300M.

**Distributed training.** Because each batch is sampled from a single session, we are able to significantly streamline memory and distributed communication overhead by allocating groups of data sessions across ranks. This allows us to partition parameters into *shared* weights (the transformer backbone) and *rank-specific* weights (i.e. identity embeddings for the sessions on each of the ranks). Accordingly, we implement a custom DDP-style training loop in which only the shared parameters are synchronized across ranks. Gradients for shared parameters are packed into flat, pre-allocated buckets and reduced asynchronously via `all_reduce`. While the collective is in flight, each rank independently clips and steps the rank-specific optimizer on its local embedding parameters, effectively overlapping communication with computation. Once the reduction completes, the averaged shared gradients are unpacked, clipped, and stepped with a second optimizer. This bucketed, communication-overlapped scheme keeps GPU utilization high even at large node counts. Our distributed training strategy achieved record #22 on the nanoGPT speedrun (Jordan et al., 2024) competition leaderboard[1], demonstrating its competitiveness beyond neuroscience applications.

**Compilation and kernel selection.** We compile the model with `torch.compile` using the `max-autotune` mode. The structured sparsity of our local sliding-window and causal attention masks is implemented via PyTorch's FlexAttention API, which compiles the block-sparse mask patterns into fused Triton kernels and avoids materializing the full attention matrix. Moreover, we specifically design the tokenization strides and local-attention window sizes for each modality to align with FlexAttention's block-size boundary of 128 tokens, ensuring that the sparse block structure incurs no wasted computation from partially occupied blocks. Similarly, we design all masking configurations

---

[1]`https://github.com/KellerJordan/modded-nanogpt`

to produce a fixed input sequence length of 4096 tokens for the multi-modal fusion transformer. This fixed shape eliminates recompilation across masking configurations and enables `torch.compile` to cache a single optimized sub-graph for the entire training run. To further minimize overhead, we are careful to cache compiled FlexAttention masks associated with each unique masking configuration.

**Numerical precision.** All training uses automatic mixed precision with BF16 for forward and backward passes, while maintaining full FP32 precision for master weights and optimizer states. We disable BF16 reduced-precision accumulation to ensure numerical stability during training.

## B  BASELINES

To establish baseline comparisons while managing computational costs, we train state-of-the-art baseline models on the smallest nested dataset containing eight mice (the seven evaluation mice plus one additional training mouse). This approach ensures that all methods are compared under identical conditions while keeping baseline training tractable. We train all baselines on 8 recordings from 8 unique mice – 5 fully released mice from the sensorium 2023 competition (keeping the original train-validation-test splits), 2 mice from the sensorium 2022 competition that were used for the test split and one session from the MiCRONs collection, which was used for training only. The same 8 mice were used in the smallest scaling experiment from Fig. 7.

### B.1  CEBRA

**CEBRA explanation:** CEBRA performs dimensionality reduction on neural activity using InfoNCE contrastive learning, where positive and negative pairs are defined by auxiliary variables such as time or behavior. When the auxiliary variable is discrete, for example a left or right wheel turn, it selects positives uniformly from all samples with the same label. When the variable is continuous, such as running speed or pupil direction, it chooses a random point within a time window around the sample and then find the closest match in the dataset using either Euclidean or cosine distance; this sample becomes the positive pair, which adds diversity and prevents repeatedly selecting the same example. Negative pairs are sampled randomly. For decoding, CEBRA encode neural responses, find the nearest latent vectors for responses in the training set, and returns their associated behavioral variables as predictions.

**Model hyperparameters:** We trained a joint model for 8 mice, using a batch size of 512 and learning rate of $3 \cdot 10^{-4}$. The network contained 256 hidden units and produced 128-dimensional outputs (both doubled relative to the Allen example `https://cebra.ai/docs/demo_notebooks/Demo_Allen.html`). Training ran for up to 50,000 iterations with cosine distance as the loss metric. The model used a temperature of 1, time-delta conditioning to enable behavior mode, and time offsets of 5. As CEBRA requires same frequencies between responses and behavior, both were resamples to 20 Hz, in order to compute correlation on the same predictions as for the OmniMouse. Please note that downsampling from 30 Hz responses is not reducing any information as responses were upsampled from 6-16 Hz to 30 Hz and the upsampling is done with nearest-neighbor interpolation.

### B.2  UNIVERSAL SPIKE TRANSLATOR

**Universal Spike Translator explanation:** The Universal Spike Translator Zhang et al. (2024) performs a self-supervised modeling approach called multi-task-masking (MtM). The model alternates between masking out and reconstructing neural activity across different time steps and neurons. It uses a learnable token that provides the model with context about the specific masking scheme that is being applied during training, allowing for "mode switching" at test time for different downstream tasks. During training, the masking schemes are sampled randomly which are: **(1) Neuron masking:** Randomly masks individual neurons and reconstructs their activity using the unmasked neurons as context. **(2) Causal masking:** Masks future time steps and predicts them using the past steps as context.

**Model hyperparameters:** We started with the default hyperparameters from "ndt1_stitching_prompting" and "ssl_session_trainer" configs from `https://github.com/colehurwitz/IBL_MtM_model` and performed a Bayesian hyperparameter search using weights & biases, `https://docs.wandb.ai/models/sweeps`. The sweep configs are available at `https://github.com/sensorium-competition/IBL_MtM_model/tree/wandb_sweep`. Please note that compared to our forecasting settings, IBL does not take behavior as model input.

### B.3  LATENT DYNAMIC MODEL

**Latent dynamic model explanation:** This is a probabilistic model that predicts the joint distribution of neuronal responses from naturalistic video stimuli and stimulus-independent latent factors. Specifically, the model predicts time-varying neuronal response using a Zero-Inflated-Gamma (ZIG) distribution to model the distribution of neuronal responses conditioned on the stimulus and the latent

factor. This is a modification of the deterministic factorized 3D convolutional core and a Gaussian readout, where we have an additional encoder that takes a subset of neurons as input to derive a latent variable. This latent variable is then combined with the transformed visual input to predict the activity of other neurons in the session. The model is trained by maximizing the Evidence Lower Bound (ELBO) of $p_{ZIG}(y|x)$ via variational inference.

**Model hyperparameters:** For both SENSORIUM 2023 baseline and Schmidt et al. (2025) baseline we used the default hyperparameters from Schmidt et al. (2025): 3 layer core with both spatial and temporal kernel = 11 in the first layer and 5 on the layer two and three. For more details see App.C from Schmidt et al. (2025). All data modalities were upsampled to 30 Hz as both SENSORIUM 2023 baseline and Schmidt et al. (2025) latent model require all modalities to have the same frequencies. Both SENSORIUM 2023 baseline and Schmidt et al. (2025) latent model predict 42 samples from a 60-frame video input, we always used only last 30 frames for evaluation, to make it consistent with OmniMouse, who was trained to predict 30 samples. Please note that OmniMouse supports flexible size of predictions, while SENSORIUM 2023 baseline and Schmidt et al. (2025) latent model has a 'burn period', e.g. cannot predict responses for the first few frames of the given video.

### B.4 POYO+

**POYO+ explanation:** POYO+ (Azabou et al., 2025) performs multi-task neural decoding by tokenizing neural activity. The tokeization follows POYO (Azabou et al., 2023) and encodes raw data into unit-level embeddings and spike times. As POYO+ is working with calcium imaging and not spikes, it also encodes the amplitude of a neuronal response not just unit-level embeddings. Its encoder uses cross-attention to keep the number of tokens fix despite potentially non-fixed input size. For flexible decoding, the model constructs query tokens by summing a learned task embedding, a session-specific embedding, and an output timestamp. It is trained end-to-end on a mixture of classification, regression, and segmentation tasks, using a weighted sum of cross-entropy and mean-squared error losses. To generate predictions, a multi-modal decoder queries the latent space via cross-attention and passes the resulting tokens through task-specific linear layers to reach the target dimensionality.

**Model hyperparameters:** We ran an extensive hyperparameter search (>100 runs) on POYO+. We used a Bayesian sweep via Weights & Biases for hyperparameter optimization; the code and sweep settings are available at `https://github.com/pollytur/torch_brain/tree/eight_mice_benchmark/examples/poyo_plus`.

## C  SUPPLEMENTAL RESULTS

**OmniMouse enables systematic evaluation of how neuronal context shapes predictive performance.** We evaluated OmniMouse on conditions not seen during training, systematically varying neuronal history duration (10-25 samples) and population context size (16-2048 neurons) for population context tasks. Performance scaled smoothly with context availability across all conditions Fig. S2. When video was available, performance plateaued more quickly for forecasting but continued to improve for population context, suggesting that nearby neurons carry complementary information beyond visual input. These systematic evaluations demonstrate that OmniMouse has learned generalizable representations of neural variability, enabling quantitative assessment of how different sources of context—temporal history contribute to explaining variability in neural responses.

Furthermore, we hypothesized that harder tasks might benefit more from scaling, as shown for large language models (Minaee et al., 2024; Naveed et al., 2025). To test this, we varied the causal context $\in [10, 15, 20, 25]$) for the forecasting tasks and population context size (*context* $\in [16, 32, ..., 1024, 2048]$), with smaller contexts being more challenging prediction tasks. We also compared performance with and without 2 seconds of video input. Fig. S2 confirms our hypothesis: performance improves consistently as context grows, hence, bigger context indicates an easier task. Non-video conditioned regimes scale more steeply. However, contrary to LLMs, in our case scaling model size does not preferentially benefit harder tasks: across all tasks, results for different model sizes do not meaningfully change for harder tasks.

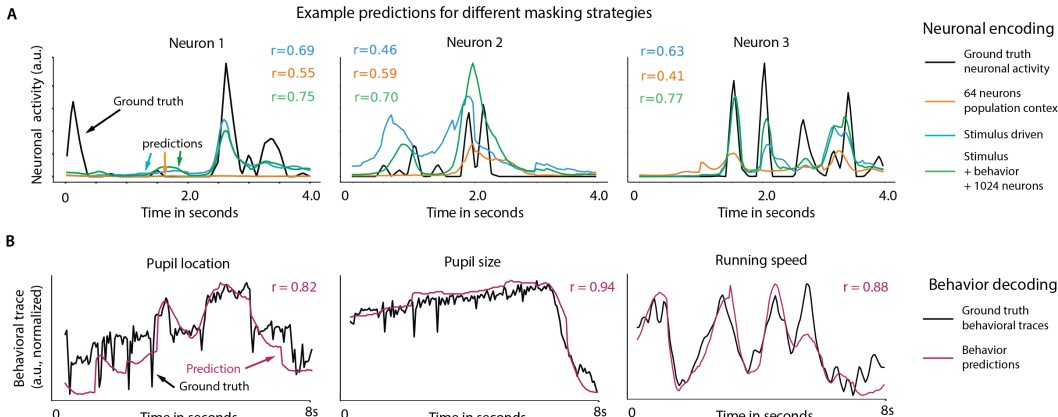

Figure S1: **Example predictions of neuronal activity and behavioral variables**. **A**. Here we show three example neurons and their ground truth neuronal activity for 4 seconds (black). We also show the model prediction of OmniMouse for three evaluation conditions: *population context of 64 neurons* (orange), *stimulus-encoding* (blue), *stimulus + behavior + neuron context* (green). The predictive performance, shown as pearson correlation *r* is increasing with more information provided to the model. Our model is designed to disentangle the relative contributions of sensory input, behavior, and population dynamics to individual neurons' activity. **B**. Ground truth and predictions for behavioral variables.

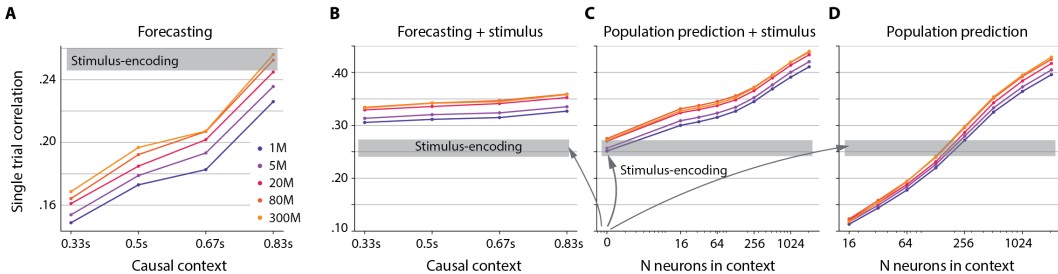

Figure S2: **Using the models capabilities to investigate context lengths for forecasting and population context tasks.** **A**. Forecasting with a change of causal context, i.e. how many samples of neuronal activity are unmasked. A causal context length of 10 corresponds to one third of a second of neuronal activity. **B**. Same change of forecasting context as **A**, but with video. **C**. Performance improvements in addition to video with population context. # neurons in context = 0 means that all neurons are masked, and the model conditions its prediction purely on the visual stimulus. In all panels, the stimulus-encoding performance is denoted by the gray box for ease of comparison. Remarkably, as seen in panel A, forecasting a whole second of neuronal activity given the past second yields a comparable predictive performance as showing the entire two seconds of the visual stimulus. **D**. Population context only, 16 - 2048 neurons

## D   SENSORIUM COMPETITION EVALUATION

We evaluate our model on the SENSORIUM 2023 competition test set, which allows direct comparison against the state of the art model of predicting mouse visual cortex responses from video stimuli. We use OmniMouse-80M, freeze the entire model, and train only the neuron and animal embeddings using the released training data of five mice provided by the competition. For SENSORIUM 2022, we report the results of OmniMouse-80M trained on the largest dataset collection, which included the training set of the two animals used in SENSORIUM 2022.

### D.1   DISTRIBUTION OF SESSIONS PER MOUSE

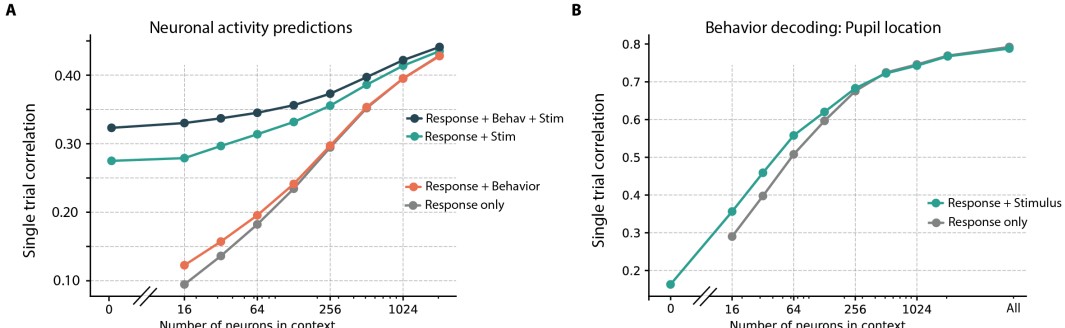

Figure S3: **Systematic evaluation across mask configurations**. We evaluate the neuronal prediction and behavior decoding performance of OmniMouse-80M by systematically varying the model inputs via masking. Only masks using N = [64, 256, 1024] have been seen during training. OmniMouse generalizes to unseen conditions, and allows to systematically study the contribution of visual stimulus, behavioral variables, and neuronal (sub)-population activity. **A**. Neuronal activity predictions given different amounts of visible neurons in context. **B**. Behavioral decoding.

Table S7: **Sensorium 2023 benchmark results.** Models with $\Sigma$ suffix denote ensemble predictions. We use n=5 models in the OmniMouse ensemble, from different random seeds, which determines model initialization and ordering of training batches. $\uparrow$ indicates higher is better. We either run the full multi-modal training, or only train the model with a single masking condition( *Unimodal*) – predicting neuronal responses conditioned on behavior and visual stimulus – comparable to all other models of the competition.

| Model | Training | Main track $\uparrow$ | OOD track $\uparrow$ |
|---|---|---|---|
| DwiseNeuro-$\Sigma$ (Turishcheva et al., 2024) | end-to-end | 0.291 | 0.221 |
| OmniMouse-5M-Unimodal | end-to-end | $0.288 \pm .003$ | $0.256 \pm .002$ |
| OmniMouse-5M-Unimodal-$\Sigma$ | end-to-end | 0.332 | 0.296 |
| OmniMouse-5M | end-to-end | $0.295 \pm .005$ | $0.263 \pm .003$ |
| OmniMouse-5M-$\Sigma$ | end-to-end | 0.327 | 0.293 |
| OmniMouse-80M | frozen | $0.313 \pm .001$ | $0.274 \pm .001$ |
| OmniMouse-80M-$\Sigma$ | frozen | 0.327 | 0.288 |

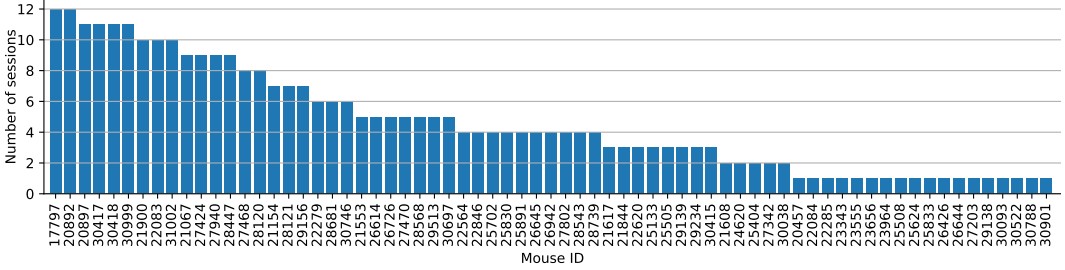

Figure S4: **Distribution of 316 sessions across 69 mice.** More than 100 sessions come from first 10 mice.

# E  DATASET DETAILS

### E.0.1  NESTED SCALING DATASET CONSTRUCTION

The nested dataset was constructed such that for the 7 mice we conducted evaluation on - 3 mice we had repeated sessions, such that the number of repeats grew proportionally to the dataset growth, and 4 other mice had a single session. As session-per-mice distribution is highly skewed, the other sessions were samples randomly.

## E.1  NEUROPHYSIOLOGICAL EXPERIMENTS

Model evaluation was performed on neurophysiological data from Sensorium 2022 ((Willeke et al., 2022), Mouse 1 and 2, evaluation animals for Sensorium and Sensorium Plus tracks) and Sensorium 2023 ((Turishcheva et al., 2024), all animals). Model training was performed on historical data, including data from MICrONS Consortium (2025), Wang et al. (2025), Ding et al. (2025b), Ding et al. (2025a), Fahey et al. (2019), Willeke et al. (2022),Turishcheva et al. (2024), but also included data not previously published.

All procedures were approved by the Institutional Animal Care and Use Committee of Baylor College of Medicine. Seventy-eight mice (Mus musculus, 32 females, 46 males, P50–155 on day of first scan) expressing GCaMP6s in excitatory neurons via Slc17a7-Cre and Ai162 transgenic lines (recommended and generously shared by Hongkui Zeng at Allen Institute for Brain Science; Jackson Labs stock 023527 and 031562, respectively) were anesthetized and a 4 mm craniotomy was made over the visual cortex of the right hemisphere as described previously (Reimer et al., 2014; Froudarakis et al., 2014). In two of the seventy-three animals, GCaMP6s was additionally expressed in inhibitory neurons via DLX5-CreER (Jackson Labs stock 010705), following treatment with tamoxifen (orogastric gavage of tamoxifen (Sigma Aldritch T5648) dissolved in corn oil (Sigma Aldritch C8267) at 15 mg/mL, 200 mg/kg body weight, two doses two days apart, second dose >= 13 days before the first included scan).

Mice were head-mounted above a cylindrical treadmill and calcium imaging was performed using Chameleon Ti-Sapphire laser (Coherent) tuned to 920 nm and a large field of view mesoscope (Sofroniew et al., 2016) equipped with a custom objective (excitation NA 0.6, collection NA 1.0, 21 mm focal length). Laser power after the objective was increased exponentially as a function of depth from the surface according to:

$$P = P_0 \times e^{(z/L_z)} \tag{3}$$

Here P is the laser power used at target depth z, P0 is the power used at the surface (typically not exceeding 25 mW), and $L_z$ is the depth constant (160-220 µm). The greatest laser output of ca. 112 mW was used at approximately 400-500 µm from the surface.

The craniotomy window was leveled with regards to the objective with six degrees of freedom. Pixelwise responses from an ROI spanning the cortical window (1.7-4 mm diameter FOV, >0.2 px/µm, superficial cortex, >2.47 Hz) to drifting bar stimuli were used to generate a sign map for delineating visual areas (Garrett et al., 2014). In some but not all cases where the imaging field of view spanned multiple areas, area boundaries on the sign map were manually annotated. Imaging FOV of varying dimensions were targeted to lie within the boundaries of visual cortex, and may span between primary visual cortex and surrounding higher visual areas depending on the scan design.

Scan dimensions typically fell into one of three categories. Local field of view scans contained multiple imaging planes at different depths (10-13 planes, most commonly with 5 µm z spacing but ranging between 3 and 45 µm z spacing), with each plane spanning 600-630 × 600-630 µm (240-252 × 240-252 pixels, 0.4 px/µm resolution ), acquired most commonly at 7.98 Hz (range 4.34-8.31 Hz). Large field of view scans contained single imaging planes at a single depth, with each plane scanning 1.5 - 3 mm diameter (0.33 - 0.4 px/um resolution), acquired at between 6.5 - 12.4 Hz. In between are scans containing multiple imaging planes at different depths (2-5 planes, with variable interplane spacing between 5 and 150 µm), with each plane spanning approximately 0.8-1.2 mm diameter (0.4-0.6 px/µm resolution), acquired at between 6.3 and 9.6 Hz. Scans with multiple planes, especially at high sampling densities (ex. 5 µm z spacing), have a high likelihood of multiple segmented traces

emerging from multiple planes intersecting with the soma of a single neuron in a single scan. Multiple scans were also often collected from the same animal, and as a result single biological neurons may be recorded across multiple scans.

Movie of the animal's eye and face was captured throughout the experiment. A hot mirror (Thorlabs FM02) positioned between the animal's left eye and the stimulus monitor was used to reflect an IR image onto a camera (Genie Nano C1920M, Teledyne Dalsa) without obscuring the visual stimulus. The position of the mirror and camera were manually calibrated per session and focused on the pupil. Field of view was manually cropped for each session to contain the left eye in its entirety, although across different experiments the field of view may have additionally contained more or less of the face, centered or not centered on the eye, or characterized the pupil at different resolutions. Video was captured at ca. 20 Hz. Frame times were time stamped in the behavioral clock for alignment to the stimulus and scan frame times. Video was compressed using Labview's MJPEG codec with quality constant of 600 and stored in an AVI file.

Light diffusing from the laser during scanning through the pupil was used to capture pupil diameter and eye movements. A DeepLabCut model (Mathis et al., 2018) was trained as previously described (Turishcheva et al., 2024) on 17 manually labeled samples from 11 animals to label each frame of the compressed eye video (intraframe only H.264 compression, CRF:17) with 8 eyelid points and 8 pupil points at cardinal and intercardinal positions. Pupil points with likelihood >0.9 were fit with the smallest enclosing circle, and the radius and center of this circle was extracted. Frames with < 3 pupil points with likelihood >0.9, or producing a circle fit with outlier > 5.5 standard deviations from the mean in any of the three parameters (center x, center y, radius) were discarded. Gaps in behavior were replaced by linear interpolations over the whole session, if there were more than 2 frames with gaps, then the video is removed.

The mouse was head-restrained during imaging but could walk on a treadmill. Rostro-caudal treadmill movement was measured using a rotary optical encoder (Accu-Coder 15T-01SF-2000NV1ROC-F03-S1) with a resolution of 8000 pulses per revolution, and was recorded at approx. 50-100 Hz in order to extract locomotion velocity.

Visual stimuli were presented with Psychtoolbox 3 in MATLAB (Brainard & Vision, 1997; Kleiner et al., 2007; Pelli, 1997) to the left eye with a $31.8 \times 56.5$ cm (height $\times$ width) monitor (ASUS PB258Q) with a resolution of $1080 \times 1920$ pixels positioned 15 cm away from the eye. When the monitor is centered on and perpendicular to the surface of the eye at the closest point, this corresponds to a visual angle of 3.8 °/cm at the nearest point and 0.7 °/cm at the most remote corner of the monitor. As the craniotomy coverslip placement during surgery and the resulting mouse positioning relative to the objective is optimized for imaging quality and stability, uncontrolled variance in animal skull position relative to the washer used for head-mounting was compensated with tailored monitor positioning on a six dimensional monitor arm. The pitch of the monitor was kept in the vertical position for all animals, while the roll was visually matched to the roll of the animal's head beneath the headbar by the experimenter. In order to optimize the translational monitor position for centered visual cortex stimulation with respect to the imaging field of view, we used a dot stimulus with a bright background (maximum pixel intensity) and a single dark square dot (minimum pixel intensity). Dot locations were randomly ordered from a grid tiling a portion of the screen, either a 10 $\times$ 10 grid tiling a central square (approx. 90° width and height, 10 repeats per location, 200-300 ms presentation at each location), or a $5 \times 8$ grid tiling the majority of the monitor (approx. 93° height and 119° width, 20 repeats per location, 200 ms presentation at each location). The final monitor position for each animal was chosen in order to center the population receptive field of the scan field ROI on the monitor, with the yaw of the monitor visually matched to be perpendicular to and 15 cm from the nearest surface of the eye at that position.

A photodiode (TAOS TSL253) was sealed to the top left corner of the monitor, and the voltage was recorded at 10 kHz and timestamped on the behavior clock (MasterClock PCIe-OSC-HSO-2 card). Simultaneous measurement with a luminance meter (LS-100 Konica Minolta) perpendicular to and targeting the center of the monitor was used to generate a lookup table for linear interpolation between photodiode voltage and monitor luminance in cd/m² for 16 equidistant values from 0-255, and one baseline value with the monitor unpowered.

At the beginning of each experimental session, we collected photodiode voltage for 52 full-screen pixel values from 0 to 255 for one second trials. The mean photodiode voltage for each trial $V_{pd}$ was fit as a function of the pixel intensity $V_{in}$:

$$V_{pd} = B + A \times V_{in}^{\gamma} \tag{4}$$

in order to estimate the $\gamma$ value of the monitor ($\approx 1.50 - 1.76$). All stimuli were shown with no $\gamma$ correction.

During the stimulus presentation, sequence information was encoded in a 3 level signal according to the binary encoding of the flip number assigned in-order. This signal underwent a sine convolution, allowing for local peak detection to recover the binary signal. A linear fit was applied to the trial timestamps in the behavioral and stimulus clocks, and the offset of that fit was applied to the data to align the two clocks, allowing linear interpolation between them. The mean photodiode voltage of the sequence encoding signal at pixel values 0 and 255 was used to estimate the luminance range of the monitor during the stimulus, with typical maximum values of approx. 10-12 cd/m².

