# OpenReview forum: "OmniMouse: Scaling properties of multi-modal, multi-task Brain Models on 150B Neural Tokens"
_ICLR.cc/2026/Conference — ICLR 2026 Poster_

### Official Review · Reviewer_8RvB · 2025-10-29

**Soundness:** 3
**Presentation:** 3
**Contribution:** 3
**Rating:** 4
**Confidence:** 5

**Summary:**

This paper presents a model architecture and training methodology and uses it to analyze data and model scaling trends for neuroscience data. It discovers positive trend in data scaling, and a limited scaling on the model side (suggesting the need for more data). The overall message is an important contribution. My main concerns are with one of the baseline used in evaluation, and some concerns about phrasing.

**Strengths:**

- Analysis of both data scaling and model scaling trends is very valuable for the community. One of the first papers to do that.
- Evaluation strategy is well designed, and a good number of tasks are included.

**Weaknesses:**

I like the overall message of this paper, but I think the problems mentioned below need to be addressed before I can recommend acceptance.

**Main remaks**
- I would prefer the reasoning for upsampling the data was put in the main text (I see it is in the appendix right now.)
- (Opinionated) Just using number of neurons to compare dataset sizes doesn't seem like a good idea. This is reference to the line: "We used a dataset of over 3 million single-unit neuronal recordings – an order of magnitude larger than...". After all, I could have 1 second long recordings of 10 million neurons, and that dataset would not be considered big. To me, it seems a metric like "neuron-hours" would be better. i.e. including both the number of neurons and the recording durations to indicate data size.
- Section 3, Data utilization: "A key novelty is our ability to sample...". Can you explain how this is novel in context to existing work? As far as I know, POYO and POYO+ have used this kind of arbitrary continuous sampling, and it has been a part of the open-source [`torch_brain`](https://github.com/neuro-galaxy/torch_brain) package for quite some time.
- My biggest issue: When comparing the performance of behavior decoding, why not compare with something like POYO (individual behaviors), or POYO+ (multi-behavior)? These models (and many other recent methods) have proven to be much better than the CEBRA baseline. Comparing with something that is not a leading method for behavior decoding diminishes the results. Including stronger baselines would help convince the reader that the suggested expensive large-scale semi-supervised training does indeed lead to better performance, better than simple supervised learning using strong methods. I understand the paper is more focused on scaling, but it is important having a better reference of where purely supervised methods are in comparison.

**Nits**
- Fig 1. Missing y-axis labels
- Some places, such as paragraph 2 of Introduction should have em-dashes `---` instead of en-dashes
- Line 071 - comma after task - "single modality, task, or dataset"
- Typo: Fig 3 caption - "tookens"
- Line 249 - "shared linear" -> "shared linear layer"

**Questions:**

**Questions**
- It is unclear whether the dataset used for pretraining is public or private.
- Can you please expand on "restart training from intermediate checkpoints every 20k steps"? (From section 4, Training paragraph) Which intermediate checkpoints do you restart from? Why restart? Why not just continue with a step change in LR?
- In section 5, Forecasting: Why "40 frames of behavior" were used to condition the forecasting? Why not 30?

---

> ### Author Response · Authors · 2025-11-21
> **Clarified details, published the code, data will be available upon publication**
>
> Thank you for appreciating the paper’s main message and for noting that our scaling analysis is *“valuable for the community”* and our evaluation strategies are *“well designed”*.
>
> As for your remarks:
> * *“Data upsampling”*: The original videos were shown to animals at 30-60 Hz (line 186) and downsampled for 30 Hz, pupil variables were recorded at 20 Hz (lines 1078-1084), the running speed was recorded at 50-100Hz (lines 1097-1100), and the neuronal responses were recorded at 6-14 Hz (line 160). For OmniMouse training, we upsampled the neuronal responses to 30Hz to be comparable with Sensorium Baselines and *downsampled* the running speed to 20Hz to align with the other behavioural variables. As suggested, we made it more clear in Sec. 3.
> * *”Neuron-hours for datasets comparison"*: We agree that neuron-hours are more informative than raw neuron count, and we report it in Fig. 2B: ~5.73×10⁶ neuron-hours for the training set. The Brain-Wide-Map dataset did not provide this metric, but we can estimate it. They recorded 621,733 neurons, with ~5 seconds of data per trial (0.7 s quiescence + 0.1 s tone + ~2 s before movement + ~2 s feedback; Fig. 1b) and 645 trials per session (see *“Behavioural task”* section). This gives: 645 trials × 5 s ≈ 0.9 hours per neuron. Thus, the total is: 621,733 neurons × 0.9 hours ≈ <1×10⁶ neuron-hours — still an order of magnitude smaller than ours.
> * *”Comparision to torch_brain”*: Thank you - we could have contextualized this statement better. It was meant from the perspective of visual cortex modeling, where periods without stimulus (blank screen) are typically excluded and datasets are organized as (stimulus, response) pairs. Our design reconstructs the full visual timeline, including inter-trial blanks, for a large number of diverse recordings. However, we understand how our statement can be misread in the broader context. We have therefore removed this framing from the manuscript.
> * *”POYO baseline”*: We are not aware of papers directly comparing POYO and CEBRA for behavior (rather than movie frame) decoding, but we understand the interest in a transformer-based comparison for behavior decoding. *We are training POYO+ on our data, for now it did not outperform CEBRA but we are still working on the hyperparameters tuning.*
> For all other tasks, to the best of our knowledge, we have compared OmniMouse to state-of-the-art methods. For the stimulus-driven condition, we compare against the leading models of the public sensorium 2023 benchmark [2] and set a new state-of-the-art, also when comparing on the same amount of training data (see reply for reviewer Lnv7). For the stimulus+response conditions, there are no established leaderboards and we chose to compare against recent work [3]. Lastly, in the response forecasting condition, we benchmark our model against IBL/MtM [1], a recent transformer architecture. We note that our comparison across several baselines is already broader than what is typically included in related work.
>
> As for your questions:
> * *”Data access”*: Thanks for raising this critical issue. We have not issued a data or code availability statement, which was an oversight. Since this is one of the largest neuroscience datasets combining visual stimuli, behavior, and visual cortex activity, preparation for the release is still ongoing. *The full dataset will also be released within the next six months.*
> Here **we share the code** for data-loading code (https://tinyurl.com/3tx9cebj) and model training (https://tinyurl.com/4s3nyksr).
> * *”Training from intermediate checkpoints”*: Thank you. To clarify: we want to evaluate the models trained till convergence given a fixed compute budget (Flops). For it we follow the warmup-stable-decay schedule (Hu et al. 2024; Wen et al. 2024), training each model for at least 300k steps with a fixed learning rate and saving checkpoints every 20k steps. After initial training, we start the decay phase to achieve convergence. We pick regular spaced checkpoints every 20k steps (e.g. 20k, 40k, 60k, … 300k). From each checkpoint, we continue training for 10k more steps with an inverse-square-root learning rate decay $\alpha$~$\frac{\alpha_0}{\sqrt{t}}$, where $t$ is the step.
> Once a checkpoint’s learning rate reaches zero, it gives a single final evaluation point for a given compute budget. In Fig. 5, each colored line represents a model trained with a constant learning rate, and each point on that line is a checkpoint whose learning rate has decayed to zero.
> We clarified this in the pdf as well.
> * *”Behaviour frames”*: see reply for remark 1. We used 40 behavioural samples because our model was trained on 2 seconds of behavior sampled at 20 Hz. The pupil signals were recorded at 20 Hz, and we downsampled the running speed to match this rate for consistency.
>
> Thank you for the detailed feedback and for helping improve the clarity of the paper. We hope our response and edits are helpful, and we are happy to answer any further questions.

---

> > ### Author Response · Authors · 2025-11-21
> > **Continued response (adding missing references)**
> >
> > References:
> >
> > [1] Zhang et al., 2024, https://arxiv.org/abs/2407.14668.
> > [2] Turishcheva et al., 2024 (NeurIPS), https://arxiv.org/abs/2407.09100.
> > [3]  Schmidt et al., 2025 (NeurIPS), https://arxiv.org/abs/2410.16136.

---

> > > ### Author Response · Authors · 2025-11-26
> > > **POYO+ baseline comparison**
> > >
> > > We have now completed the POYO+ experiments described in our initial response. We ran an extensive hyperparameter search (\>100 runs) on POYO+ using the same data that we trained our and all baselines on, to compare performance against CEBRA and OmniMouse. We used a Bayesian sweep via Weights & Biases for hyperparameter optimization; the code and sweep settings are available here: https://anonymous.4open.science/r/torch_brain_omnimouse/examples/poyo_plus/sweep.yaml
> > >
> > > Initial architectural search did not outperform CEBRA. We therefore also tuned the neuron sampling augmentation, as our dataset has substantially more neurons per session (\~7000) compared to \~300 neurons in the Allen dataset used in POYO+.
> > >
> > >
> > >
> > > The full results are shown in the table below:
> > >
> > >
> > >
> > > **Behavior decoding comparison**
> > > | Model | Gaze | Pupil | Running |
> > > | :--- | :-: | :-: | :-: |
> > > | **CEBRA** (Schneider et al., 2023)  | 0.52 | 0.55 | **0.51** |
> > > | **POYO+** (Azabou et al., 2025)  | 0.56 | 0.63 | 0.47 |
> > > | **OmniMouse-5M ( data-matched)** |**0.68** |**0.66** |0.44 |
> > > | **OmniMouse-1M** | 0.75| 0.73| 0.55 |
> > > | **OmniMouse-5M**  | 0.76| 0.74| 0.57|
> > > | **OmniMouse-20M** |0.78 | 0.75| 0.73|
> > > | **OmniMouse-80M**  |**0.80**|**0.76** |**0.75** |
> > > | **OmniMouse-300M**  |**0.80** |**0.76** |0.73|
> > >
> > >
> > > In data-matched conditions, OmniMouse outperforms both CEBRA and POYO+ on gaze and pupil size decoding, while performing slightly weaker on running speed. We suspect this gap arises because the running signal is sparse—mice often do not run—and CEBRA's contrastive loss may be particularly effective at separating running vs. non-running periods.
> > > With access to more training data, OmniMouse improves considerably across all three conditions, consistent with the scaling trends we report in the main paper.
> > >
> > > Please also refer to our response to reviewer **Lnv7** for additional data-matched baseline experiments.

---

### Official Review · Reviewer_Cuf1 · 2025-10-30

**Soundness:** 3
**Presentation:** 3
**Contribution:** 3
**Rating:** 6
**Confidence:** 3

**Summary:**

This paper studies whether the scaling laws in AI also apply to modeling neural activity. Using a dataset with 3.3 million neurons from the mouse visual cortex (78 mice, 323 sessions, and over 150 billion neural tokens), the authors trained multi-modal, multi-task transformer models ranging from 1M to 300M parameters. These models can perform neural prediction, behavioral decoding, and neural forecasting. Empirically, the authors found that performance scales with data quantity but saturates with increasing model size, indicating that current modeling efforts in neuroscience are data-limited.

**Strengths:**

This paper makes a timely contribution by systematically studying scaling laws in neurofoundation models using an unprecedentedly large dataset. The work is novel in introducing the first large-scale, multi-modal, multi-task transformer that unifies neural encoding, decoding, and forecasting with naturalistic video inputs. The paper is also well written and organized.

**Weaknesses:**

1. The paper is impressive in scale, but several aspects could be improved. The contribution is empirical rather than methodological. The proposed transformer largely builds on existing model designs (e.g., POYO+ and prior multi-modal fusion techniques), with limited architectural innovation beyond scaling and integration. Clarifying which components are novel vs. adapted from prior work would help better explain the contribution.

2. The paper would benefit from more qualitative analyses or visualizations to show what the model has learned.

3. Although the inclusion of naturalistic video inputs is interesting, the current analyses do not disentangle how much information comes from visual vs. behaviors (e.g., pupil location, size, running speed).

**Questions:**

1. The single-trial correlation metrics in Fig. 5 are quite low. Could the authors clarify the reason for this? For example, is it due to the use of naturalistic video stimuli rather than repeated trials? Also, why was correlation chosen as the primary evaluation metric, given that it only captures linear relationships and does not account for nonlinear dependencies or variance structure in the data?

2. The paper claims that scaling in neuroscience is limited by data rather than model size, unlike in AI. In Fig. 5, it seems that as data increases, model size should also grow; otherwise, performance may saturate. This implies that model scaling is still necessary when handling larger datasets. In AI, there is already abundant data, which may make increasing model size particularly helpful. I am curious if the authors have considered this factor.

---

> ### Author Response · Authors · 2025-11-21
> **Added requested analysis and clarified contributions**
>
> Thank you for recognizing our *“timely contribution by systematically studying scaling laws in neurofoundation models”*.
>
> As for weaknesses mentioned:
> * *”Contributions”*: Thank you for the opportunity to clarify our contributions. While building on POYO tokenization and MtM [9] masking, we extend both. Our first contribution is explicitly integrating sensory (visual) input, which we tokenize and encode using the Hiera architecture [8] (which is novel in the context of encoding visual stimuli). Second, we extend the POYO-style single-neuron tokenization also to the decoder, allowing us to (a) flexibly mask individual neurons in both encoding and decoding, and (b) make masking more flexible: the model can distinguish between missing activity and intentionally masked information. In our model, individual neurons can be selectively masked or unmasked during both training and inference, while MtM tokenization relies on a fixed shape vector of shape (1 $\times$ neurons), and zero could mean both no activity or masked activity. In our case we do not provide any tokens for masked neurons. Taken together, it’s possible in our model to precisely control which neurons the predictions are conditioned on. This combination of single-neuron resolution (which by itself is not novel, see POYO) and multi-task model with masking (which has also been done, see NEDS, albeit without video) is what has been missing and what our model contributes.
> * *”Qualitative analyses”*: We agree that visualizing model predictions helps to understand what the model has learned. We added a new supplemental figure S1 with example predictions across different evaluation regimes.
> *Could you clarify which qualitative analysis you mean, if this is not sufficient?*
> * *”Separating information between visual and behavioral input”*: Thanks for raising this point. Our model naturally disentangles information from different modalities. We ran an experiment using different combinations of inputs to predict neurons never used as input in this experiment (>7k neurons). This analysis shows that behavior alone explains little variance (1% correlation), likely because this type of masking was not used during training. A 32-neuron population context predicted 13% correlation; adding behavior to it improved a bit (+2%). The stimulus was the strongest individual predictor, resulting in 28% correlation. Interestingly, using behavior as an additional input improved predictions more (+4%) than adding the 32-neuron population context (+1%). This could be because neuronal responses in the visual cortex are driven by the stimulus. Combining all three modalities gives the best predictions (+5%), showing our model effectively integrates stimulus-driven and latent-state components.
> |Input|Pearson correlation|seen in training|
> |:-|:-|:-|
> |Behavior only|0.01|✘|
> |32 population context |0.13|✘|
> |32 population context + behavior|0.15|✔️|
> |Stimulus only|0.28|✔️|
> |Stimulus + 32 population context|0.29|✘|
> |Stimulus + behavior|0.32|✔️|
> |Stimulus + 32 population context + behavior|0.33|✔️|

---

> > ### Author Response · Authors · 2025-11-21
> > **Continued response**
> >
> > As for your questions:
> > * *”Low correlation values and other metrics”*: Although the correlations may appear low, this represents state-of-the-art performance for neural prediction tasks, as seen in benchmarks such as the Sensorium 2023 competition [3]. Neural responses to naturalistic video stimuli are inherently noisy and variable, making single-trial predictions especially challenging. The low single-trial correlation therefore reflects this biological variability, not a weakness of our model.
> > We use correlation as the main evaluation metric because it is standard in the field [4] and directly comparable to prior work [3]. It provides an intuitive measure of how closely the predicted responses follow the observed neural activity.
> > A variance-based metric would be ideal but is hard to define and remains an open research problem. Metrics like FEVE [5] require repeated trials to be estimated and thus only capture stimulus-driven variance, ignoring trial-to-trial variability, and cannot be estimated when behavioral data are included, as behavior never repeats exactly [6]. Similarly, while $CC_{norm}$ [7] accounts for intrinsic variability, it also  requires repeated presentations of the same input which is not always present in the data.
> > * *”Data-limited regime claim”*: We believe there is a misunderstanding - **in Fig. 5, we kept the data identical across all model sizes and only scaled the model**.
> > All models were trained once on the full dataset. The number of unique tokens was identical for every model size. Each point in Fig. 5 corresponds to a fixed number of steps. Our claim is the following: in existing neuroscience datasets, increasing model size beyond a moderate scale offers little improvement—data availability is the main bottleneck. The 20M (pink), 80M (orange), and 300M (yellow) models reach close results in panels A-B and almost the same loss and correlation, especially on the tasks with the stimuli (panels C–E). This shows that increasing parameters by 15× provides minimal gain, supporting the data-limited interpretation.
> >
> > References:
> > [1] Niell & Stryker 2010 (Neuron) doi: 10.1016/j.neuron.2010.01.033
> > [2] Stringer et al. 2019 (Science), doi: 10.1126/science.aav789
> > [3] Turishcheva et al. 2024 (NeurIPS) doi: 10.52202/079017-3777
> > [4] Pospisil & Bair 2021 (PLoS computational biology) doi: 10.1371/journal.pcbi.1009212
> > [5] Cadena et al. 2019 (PLoS computational biology) doi: 10.1371/journal.pcbi.1006897
> > [6] Willeke et al 2023 (PMLR) https://proceedings.mlr.press/v220/willeke23a.html
> > [7] Wang et al. 2025 (Nature), doi: 10.1038/s41586-025-08829-y
> > [8] Ryali et al. 2023 (ICML), https://arxiv.org/abs/2306.00989
> > [9] Zhang et al., 2024 (NeurIPS), https://arxiv.org/abs/2407.14668

---

> > ### Comment · Reviewer_Cuf1 · 2025-11-21
> >
> > Thank you for the response and for addressing my questions. The added single-neuron qualitative analyses are helpful. I was also curious whether the model can attribute which patterns in the visual stimuli drive single-neuron responses. Have the authors explored this aspect, or is the current model capable of such analyses?

---

> ### Author Response · Authors · 2025-11-21
>
> Thanks a lot for this question. Models such as ours have been used previously to investigate visual selectivity of neurons. For example, generating stimulus patterns, which maximally drive single-neuron activity or optimize for other properties, such as invariances [1], is an established and experimentally verified procedure for visual encoding models [1-8]. These stimuli are often called most exciting images (MEIs) because they are synthesized from a trained encoding network (like ours) to drive the activity of real neurons maximally under constrained optimization. For example Ding et al. [1] applied the MEI approach and found neurons in mouse area V1 and found new forms of single cell invariances, that might play a role in object segmentation by detecting boundaries based on spatial frequency. In another example, Franke et al. [7] showed that color selectivity of neurons in mouse visual cortex is changing with behavior. Our model can also be used within this framework and – because it has a higher predictive performance compared to previous models – we expect it to produce meaningful videos or images. In addition, the ability to condition the model on other modalities will allow us to study MEIs or other neuronal response properties in more detail. However, it is important to note that any novel biologically meaningful claims that we could derive from our model would require an in vivo validation, which goes beyond the scope and goal of the current work.
>
> The focus of the current paper is on model scaling, less on how it can be used for scientific discovery. But since this is one of our main long-term motivations for training this model, we’ll add a paragraph in the discussion section on how our model can be used for scientific insight.
>
>
> *References*:
> [1] Zhiwei Ding, et al., “Bipartite invariance in mouse primary visual cortex” (2023), https://www.biorxiv.org/content/10.1101/2023.03.15.532836v2.
> [2] Walker et al. 2019 (Nature neuroscience), doi: 10.1038/s41593-019-0517-x.
> [3] Bashivan, Pouya, Kohitij Kar, and James J. DiCarlo. "Neural population control via deep image synthesis." Science 364.6439 (2019): eaav9436.
> [4] Ponce, Carlos R., et al. "Evolving images for visual neurons using a deep generative network reveals coding principles and neuronal preferences." Cell 177.4 (2019): 999-1009.
> [5] Willeke, Konstantin F., et al. "Deep learning-driven characterization of single cell tuning in primate visual area V4 supports topological organization." biorxiv (2023): 2023-05.
> [6] Vystrčilová, Michaela, et al. "Interpreting convolutional neural networks to study wide-field amacrine cell inhibition in the retina." bioRxiv (2025): 2025-09.
> [7] Franke, Katrin, et al. "State-dependent pupil dilation rapidly shifts visual feature selectivity." Nature 610.7930 (2022): 128-134.
> [8] Tong et al., “The feature landscape of visual cortex” (2023) https://www.biorxiv.org/content/10.1101/2023.11.03.565500v1.

---

> > ### Comment · Reviewer_Cuf1 · 2025-11-25
> >
> > Thank you for addressing my questions. I believe the work still makes a meaningful contribution to the field, and I am leaning toward an accept. I will keep my original score of 6.

---

> > > ### Author Response · Authors · 2025-11-25
> > >
> > > Thank you for your continued engagement with our work and for recognizing that it makes a meaningful contribution to the field. To ensure we've fully addressed your evaluation, could you let us know whether our rebuttal resolved the original weaknesses you identified (architectural novelty, qualitative analyses, and modality disentanglement)? If any concerns remain, we would be glad to provide additional clarification or revisions.

---

### Official Review · Reviewer_Lnv7 · 2025-10-30

**Soundness:** 3
**Presentation:** 3
**Contribution:** 3
**Rating:** 6
**Confidence:** 3

**Summary:**

This paper presents OmniMouse, a multi-modal, multi-task transformer model trained to predict activity in the mouse visual cortex. The model is trained on a new, massive-scale dataset of 3.3 million neurons from 323 recording sessions, totaling over 150 billion neural tokens.

The primary contribution is a systematic study of scaling laws, training models from 1M to 300M parameters. The authors find that performance gains saturate with increasing model size but scale reliably and consistently with increasing dataset size. This leads to the central conclusion that current models of the visual cortex are data-limited, not parameter-limited, a finding that inverts the standard scaling narrative in mainstream AI. The largest model achieves new state-of-the-art performance on several predictive tasks, including neural forecasting, stimulus-driven prediction, and behavioral decoding.

**Strengths:**

1. **Scaling study for brain foundation model**: To my knowledge, this is the first work to systematically apply the scaling laws methodology to a single-neuron, multi-modal brain foundation model of this magnitude. The primary finding is that the model performance is data-limited rather than parameter-limited, which provides a clear directive for future progress, emphasizing the need for larger and more diverse datasets.
2. **Dataset and model contribution:** The assembly of a 150B+ token dataset from 3.3M neurons is a major contribution in itself. Furthermore, OmniMouse-300M outperforms strong, specialized baselines (e.g., IBL, CEBRA) across the Sensorium 2023 competition even with its core weights frozen (training only neuron/animal embeddings).
3. **Flexible and generalizable prediction:** The model's multi-task design, based on flexible masking of modalities, is effective. The paper shows that this design allows the model to learn representations that generalize to contextual variations not seen during training.

**Weaknesses:**

1. **Datasets scale misalignment in baseline comparison:**  The OmniMouse-300M model was trained on the full 323-session dataset, whereas all baselines were trained only on the smallest 8-mice data collection to reduce computational cost. It is unclear how much of OmniMouse's SOTA performance is due to its superior architecture versus simply having access to ~40x more data than the baselines it's compared against.
2. **Generalization bottleneck on per-neuron identity embeddings:** The use of per-neuron (and per-session/animal) identity embeddings limits the generalization of the model. As shown in Table 1, these neuronal parameters account for the vast majority of the model's total parameters (e.g., 779M $p_N$ vs. 348M $p_M$ for the "300M" model). The model cannot perform zero-shot prediction on a new, unseen neuron or animal; it must be fine-tuned. A true foundation model should learn a general representation of a neuron rather than memorizing 3.3 million specific instances.

**Questions:**

1. To provide a fair architectural comparison, what is the performance of a smaller OmniMouse model (e.g., OmniMouse-80M) when trained only on the 8-mouse dataset? How does that model compare to the baselines (IBL, CEBRA, etc.) trained on the same 8-mouse dataset? This would isolate the architectural contribution from the data-scaling contribution.
2. Have you explored neuron-agnostic tokenization strategies? For example, could a neuron be represented by an embedding derived from its anatomical coordinates, its functional tuning properties (e.g., a pre-computed receptive field), or its relative position to other neurons, rather than a unique learned ID?

---

> ### Author Response · Authors · 2025-11-21
> **Added comparison on the same data, commented on zero-shot learning idea**
>
> Thank you for a thoughtful review, appreciating the novelty of our *systematic scaling law approach*  and our effective model's multi-task design.
>
> Here, we address the questions and weakness that you have raised:
> * *Baseline comparison on 8 mice*: We agree that comparing against an 8-mouse model is the appropriate one. Here we show the performance of OmniMouse-5M trained only on the same 8-mouse dataset used for all baselines. In all cases except treadmill speed decoding, our model outperforms all specialized baselines.
> **Tab.1: OmniMouse on 8 mice**
> |Prediction task|Baseline|Baseline score $\uparrow$|OmniMouse-5M (8 mice) ↑|OmniMouse-300M ↑|
> |:-|:-|:-|:-|:-|
> |Forecasting|IBL (Zhang et al., 2024)|0.12|**0.18**|**0.25**|
> |Forecasting + stim|Latent Model (Schmidt et al., 2025)|0.18| **0.26**|**0.36**|
> |Population context|N.A.|$X$|**0.27**|**0.30**|
> |Population context + stim|Latent Model (Schmidt et al., 2025)|0.16|**0.27**|**0.37**|
> |**Behavior decoding:**|||||
> |Pupil location|CEBRA (Schneider et al., 2023)|0.52|**0.68**|**0.80**|
> |Pupil size|CEBRA (Schneider et al., 2023)|0.55|**0.66**|**0.76**|
> |Running speed|CEBRA (Schneider et al., 2023)|**0.51**|0.44|**0.73**|
>
>   For Sensorium 2023 – an established benchmark with a held-out test set – the competition winners were trained on 10 publicly released mice [1], hence, we use the same 10-mice training data for OmniMouse. Following the competition winners, we also ensemble our model. Even without ensembling, our model outperforms the competition winner, and ensembling further widens the gap. Lastly, we additionally analyzed the out-of-distribution (OOD) competition track and OmniMouse again outperforms prior models.
>   **Tab.2: Sensorium 2023 Benchmark**
>   |Model|Training|Main track ↑|OOD track ↑|
>   |:-|:-|:-|:-|
>   |DwiseNeuro-Ensemble[1]|end-to-end|0.29|0.22|
>   |*OmniMouse-5M*|end-to-end|**0.30**|**0.26**|
>   |*OmniMouse-5M-Ensemble*|end-to-end|**0.32**|**0.28**|
> * *Neuron-agnostic tokenization*: This is an great comment that we’d like to elaborate on it.
> Zero-shot neuronal encoding is very challenging. While we agree that a true foundation model of mouse visual cortex should predict responses of unseen neurons, our goal here is to take a step toward such a model—not claim to have fully achieved it (now clarified in the discussion). At present, zero-shot inference on unseen data would require substantial additional effort beyond the scope of this work.
> That said, we hypothesize that a zero-shot variant may be feasible. One could show 1–10 seconds of multimodal context (behavior, visual input, full-population activity) and train a small auxiliary network to predict neuron embeddings, and use these predicted embeddings within the frozenOmniMouse model. After inferring embeddings for a new session, the model could then run inference without any fine-tuning. Designing such embedding predictors—and appropriate embedding spaces (e.g., low-dimensional VAEs)—is an important direction for future work.
> We also emphasize that *per-neuron embeddings are a feature, not a limitation*. Our long-term goal is to enable biological discovery through in-silico experiments, in the spirit of Walker et al., 2019 [2]. One of the interesting questions for it is if functional neuron subtypes exist and how to identify them. Recent work [3,4] has begun exploring this with per-neuron embeddings, but these are typically learned independently, limiting information sharing (except for the clustering loss in [5]). In contrast, our model embeds neurons at the input and allows them to interact through attention, enabling neurons to learn from each other, similar to Avora et al., 2025 [6]. This may yield richer and more meaningful functional representations, though this exploration is beyond the scope of the present work.
> Finally, we also want to note that tuning is the current de facto standard method for neuronal “prompting” (see e.g. Wang et al. 2025, POYO). The only predictive model we know that avoids fine-tuning is [6], which uses variational inference to predict the readout from stimulus–response samples. However, if the dataset for a novel neuron is larger, they do not outperform fine tuning. Extending this approach for our multimodal framework with video stimuli is a promising research direction but out of scope for this work.
>
> We hope this response clarifies our method and architectural choices, and we welcome any follow-up questions.
>
>
> References:
> [1] Turishcheva et al. 2024 (NeurIPS) doi: 10.52202/079017-3777
> [2] Walker et al. 2019 (Nature neuroscience), doi: 10.1038/s41593-019-0517-x
> [3] Ustyuzhaninov, et al. 2022 (BioRxiv), doi: 10.1101/2022.02.10.479884
> [4] Nellen et al. "Learning to cluster neuronal function." NeurIPS 2025
> [5] Arora et al. "Know Thyself by Knowing Others: Learning Neuron Identity from Population Context." NeurIPS 2025
> [6] Cotton et al. "Factorized neural processes for neural processes: K-shot prediction of neural responses." NeurIPS 2020

---

### Official Review · Reviewer_SAAW · 2025-10-31

**Soundness:** 2
**Presentation:** 3
**Contribution:** 3
**Rating:** 6
**Confidence:** 3

**Summary:**

The authors present a scaling analysis using a large neural model. The model is trained on multi modal data from head-fixed mice in a VR task consisting of visual stimulus, locomotion speed and pupil features recording. The neural data is calcium imaging confined to the visual cortex, for an overall tally of 3M neurons on 78 subjects. The model is based previous work POYO, with an additional hierarchical vision encoder to input the visual stimulus data. The authors present learning curves as a function of model size (measured in parameters) and show that performance saturates with model size, but does not saturate with input dataset size. The model performance is evaluated on a 2023 decoding benchmark with scores beating the baselines in all categories.

**Strengths:**

The scale analysis is a welcome contribution to the field.
- The discussion correctly assesses limitations of the current approach, namely the narrow brain region and behavior protocol.
- Despite the narrow brain region and behaviour type studied, the authors manage to show that their model is still data limited, which makes the claim that dataset size is a limiting factor even more compelling.

**Weaknesses:**

- The performance is compared to baselines that may not be state of the art models. For example in behaviour decoding tasks, to justify the claim of SOTA we may be more interested in others large transformer based ANNs results than in the contrastive learning CEBRA method
- No clear statement about sharing of code, models and private data for others to address the point above or even reproduce the proposed results, while the authors have made heavy use of public resources.


Minor comments:
- figure 1bc: label y-axis missing (loss)
- Code / model / data availability statement

**Questions:**

- What reasonable steps could be taken to reproduce those results from a third-party ?

---

> ### Author Response · Authors · 2025-11-21
> **We  published code, data will be available upon publication**
>
> We thank you for your positive review and the highlighting of the contribution of our scaling analysis.
>
> Please find our point-by-point response below:
> * *Additional baselines*: We understand the concern that CEBRA is not a recent large scale transformer model such as NEDS, POYO, or POYO+ [1-3]. As these papers did not directly compare behavioral decoding accuracy against CEBRA, we agree that we should show a transformer-based baseline as well to substantiate SOTA claims.
> *We are therefore training POYO+ on our data, for now it does not outperform CEBRA but we are still working on the hyperparameters tuning.*
> For all other baselines, to the best of our knowledge, we have compared OmniMouse to state-of-the-art methods. For the stimulus-driven condition, we compare against the leading models of the public sensorium 2023 benchmark [4] and set a new state-of-the-art, also when comparing on the same amount of training data (see reply for reviewer Lnv7). For the stimulus+response conditions, there are no established leaderboards and we chose to compare against recent work [5]. Lastly, in the response forecasting condition, we benchmark our model against IBL/MtM [1], a recent transformer architecture. We note that our comparison across several baselines is already broader than what is typically included in related work.
>
> * *Reproducibility*: Thank you for raising this critical issue.  We indeed have not included a data or code availability statement, which was an oversight. Since this is one of the largest datasets thus far recorded in neuroscience, and the largest of its kind with respect to rich naturalistic visual stimulation, behavioral recordings, and neuronal responses from visual cortex, the preparation for this release is still ongoing. *The full dataset will also be released within the next six months.*
> Here **we share the full code** for data-loading (https://tinyurl.com/3tx9cebj) and the code for our multi-modal model including training,evaluation, fine-tuning, and inference scripts(https://anonymous.4open.science/r/unraveling-70BA/).
>
> We thank you once again for constructive feedback and helping us to improve our paper and make it more reproducible. We welcome any follow-up questions or opportunities to clarify our contribution.
>
> References:
> [1] Zhang et al. "Neural encoding and decoding at scale." ICML 2025
> [2] Azabou et al. "A unified, scalable framework for neural population decoding." NeurIPS 2023
> [3] Azabou et al. "Multi-session, multi-task neural decoding from distinct cell-types and brain regions." ICLR 2024
> [4] Turishcheva et al. "Reproducibility of predictive networks for mouse visual cortex." NeurIPS 2024
> [5]  Schmidt et al. “Modeling Dynamic Neural Activity by combining Naturalistic Video Stimuli and Stimulus-independent Latent Factors” NeurIPS 2025.

---

> > ### Author Response · Authors · 2025-11-26
> > **POYO+ baseline comparison**
> >
> > We have now completed the POYO+ experiments described in our initial response. We ran an extensive hyperparameter search (\>100 runs) on POYO+ using the same data that we trained our and all baselines on, to compare performance against CEBRA and OmniMouse. We used a Bayesian sweep via Weights & Biases for hyperparameter optimization; the code and sweep settings are available here: https://anonymous.4open.science/r/torch_brain_omnimouse/examples/poyo_plus/sweep.yaml
> >
> > Initial architectural search did not outperform CEBRA. We therefore also tuned the neuron sampling augmentation, as our dataset has substantially more neurons per session (\~7000) compared to \~300 neurons in the Allen dataset used in POYO+.
> >
> >
> >
> > The full results are shown in the table below:
> >
> >
> >
> > **Behavior decoding comparison**
> > | Model | Gaze | Pupil | Running |
> > | :--- | :-: | :-: | :-: |
> > | **CEBRA** (Schneider et al., 2023)  | 0.52 | 0.55 | **0.51** |
> > | **POYO+** (Azabou et al., 2025)  | 0.56 | 0.63 | 0.47 |
> > | **OmniMouse-5M ( data-matched)** |**0.68** |**0.66** |0.44 |
> > | **OmniMouse-1M** | 0.75| 0.73| 0.55 |
> > | **OmniMouse-5M**  | 0.76| 0.74| 0.57|
> > | **OmniMouse-20M** |0.78 | 0.75| 0.73|
> > | **OmniMouse-80M**  |**0.80**|**0.76** |**0.75** |
> > | **OmniMouse-300M**  |**0.80** |**0.76** |0.73|
> >
> >
> > In data-matched conditions, OmniMouse outperforms both CEBRA and POYO+ on gaze and pupil size decoding, while performing slightly weaker on running speed. We suspect this gap arises because the running signal is sparse—mice often do not run—and CEBRA's contrastive loss may be particularly effective at separating running vs. non-running periods.
> > With access to more training data, OmniMouse improves considerably across all three conditions, consistent with the scaling trends we report in the main paper.
> >
> > Please also refer to our response to reviewer **Lnv7** for additional data-matched baseline experiments.

---

### Public Comment · ~Cole_Lincoln_Hurwitz1 · 2025-11-13
**Factual Errors in Literature Review and Unfair Baseline Comparisons**

As someone who has worked on similar neural foundation models, I need to flag multiple factual errors in the literature review and methodologically problematic baseline comparisons that misrepresent both prior work and this paper's actual contributions.

## Issue 1: Overstated Novelty Regarding Multi-Modal Models

**Lines 60-61:** *"But single-neuron resolution, multi-modal foundation models are still missing."*

This claim is inaccurate. **Neural Encoding and Decoding at Scale (NEDS; Zhang et al., 2025)** already presents a transformer trained at single-neuron resolution across data from 70 animals that performs:
- **Encoding**: behavior → neural activity
- **Decoding**: neural activity → behavior
- **Multi-task masking**: alternating between neural, behavioral, within-modality, and cross-modality masking

NEDS uses Poisson loss for neural responses and MSE for continuous behavior decoding—**the same multi-modal loss structure employed here**. Stating such models are "still missing" overlooks directly comparable prior work.

The meaningful distinction is that **OmniMouse adds video** as a third modality (neural + behavior + video), which NEDS does not include, but the framing should acknowledge the existence of multi-modal single-neuron models rather than claiming to pioneer this space entirely.

## Issue 2: Citation Misattribution

**Lines 133-134:** *"Jiang et al. (2025) questioned their applicability, analyzing the NDT-based model of Zhang et al. (2025), which predicted ∼30,000 spiking neurons across 74 sessions."*

This citation is incorrect. Jiang et al. (2025) analyzed **Zhang et al. (2024)** (*"Towards a Universal Translator for Neural Dynamics"*), not Zhang et al. (2025) (NEDS). This conflates two distinct models and misattributes the analysis done by Jiang et al. (2025).

## Issue 3: Overstated Novelty of Temporal Sampling Strategy

**Lines 187-192:** *"A key novelty is our ability to sample arbitrary 2-second windows from any point in the experiment, including inter-trial intervals and blank screens."*

While this is a useful capability, characterizing it as a "key novelty" is incorrect. **POYO** (Azabou et al., 2023) introduced tokenization that removes "the need for time-window binning," and allows for sampling arbitrary windows throughout experimental sessions, including inter-trial periods.

## Issue 4: Unacknowledged Prior Architectural Patterns

**Lines 295-297:** "Finally, the outputs of the decoder cross-attention block for each modality are routed to modality-specific linear readouts, projecting from dM back to the original dimensionality."

**POYO+** (Azabou et al., 2025) already implements this design pattern: *"All tokens regardless of the task are processed through the same cross-attention layer, and a custom router groups together tokens from the same task and feeds them to a task-specific linear decoder."* Presenting it without acknowledging prior use in POYO+ misrepresents the novelty of the design.

## Issue 5: Inadequately Disclosed Training Data Disparity

**Table 2** compares OmniMouse-300M (trained on **323 sessions**) against baselines trained on only **8 sessions** (lines 354-357: *"We train all state-of-the-art baselines on...eight mice...to reduce computational cost"*).

This represents a **40× difference in training data** that is not clearly indicated in Table 2's caption or prominently discussed when interpreting results. A reader examining Table 2 cannot readily determine how much of the performance gap is attributable to the model versus the substantially larger training set.

## Issue 6: Questionable Baseline Configuration for Different Recording Modality

**Appendix C.2:** The paper uses IBL (Zhang et al., 2024) as a baseline for forecasting, stating: *"We used the default hyperparameters from 'ndt1_stitching_prompting' and 'ssl_session_trainer' configs from https://github.com/colehurwitz/IBL_MtM_model"*

While the authors did train the IBL on their calcium imaging data, they used hyperparameters designed for **Neuropixels electrophysiology data** without any adaptation for **two-photon calcium imaging**. A fair comparison would require hyperparameter tuning the IBL model on calcium imaging data or at least acknowledging that the baseline may be disadvantaged by the hyperparameter mismatch.

Overall, **there is little discussion of baseline hyperparameter tuning** in this work which raises questions about the SOTA claims.

## Issue 7 - Mischaracterizing NDT3

**Lines 136-138** - The paper states that Ye et al. (2025) (NDT3) was trained on EEG data but it was trained on **intracortical recordings**.

---

## Summary

This paper makes genuine contributions: the systematic scaling study across 150B neural tokens, the empirical finding that performance is currently data-limited rather than compute-limited, and strong results across diverse prediction tasks are all valuable advances. However, these contributions need to be situated accurately within existing work.

---

> ### Author Response · Authors · 2025-11-21
> **Novelty clarifications and citations refinements**
>
> Dear Cole,
> Thank you for reviewing our manuscript and for your feedback. We appreciate the opportunity to clarify how our work relates to prior efforts in this rapidly evolving area. As neural foundation models advance rapidly, it is encouraging to see several recent papers and groups exploring related ideas. We will expand the related-work section to better describe these connections and to clarify how our contributions build on and extend this growing body of work.
> Please see our point by point responses below.
> 1. *Overstated Novelty for Multi-Modal Models*: Thanks for pointing this out. We will rephrase this sentence to avoid being misunderstood as an overly broad claim. To clarify what we meant: The emphasis was meant to be on *single-neuron*. In our model, individual neurons can be selectively masked or unmasked during both training and inference. That means you can control precisely which neurons the predictions are conditioned on. As far as we understand it, both NEDS (Zhang et al., 2025) and the MTM model (Zhang et al., 2024) tokenize neural activity at the population level, i.e. the entire population of shape (1 $\times$ neurons) is represented as a single input/output token. Could you comment on whether NEDS could forecast the activity of an arbitrary subset of neurons based on the past activity of a different subset of neurons (both subsets determined at test time)? Or could it predict behavior from a specific subset of neurons instead of all neurons in a time bin? This combination of single-neuron resolution (which by itself is not novel, see POYO) and multi-modal (which has also been done, see NEDS, albeit without video) is what has been missing and our model provides. We will also make sure in the revised manuscript to properly credit NEDS as a multimodal multitask model.
> 2. *Citation Misattribution*: Thank you. We have corrected it.
> 3. *Temporal sampling:* Thank you. We could have contextualized this statement better. It was meant from the perspective of visual cortex modeling, where periods without stimulus (blank screen) are typically excluded and datasets are organized as (stimulus, response) pairs. Our design reconstructs the full visual timeline, including inter-trial blanks, for  a large number of diverse recordings. However, we understand how our statement can be misread in the broader context. We have therefore removed this framing from the manuscript.
> 4. *POYO+ recognition*: We did not intend to claim any novelty for this architectural part but merely describe the architecture here. To emphasize that we now have added an explicit citation to POYO+ (Azabou et al., 2025): *"Finally, similar to POYO+ (Azabou et al., 2025), the outputs of the decoder..."*.
> 5. *Comparison on same data*: We agree that the training data difference between OmniMouse-300M (323 sessions) and the baselines (8 sessions) could be made clearer in Table 2. We would not call it “inadequate,” though, as we do explain it in the text (lines 353-354). Our primary claim is about data scaling behavior, not about architecture (see abstract and contribution statements). Scaling all of the baselines to the full dataset would be an enormous engineering challenge and require tremendous amounts of compute. We therefore opted to train them with a typical amount of data from prior work and tried to carefully avoid any claims that our architecture is superior to others.
> We will make the data differences more explicit in the table caption to avoid any ambiguity. To facilitate direct comparison, we are also adding results for OmniMouse trained on the same 8-session subset used for the baselines (see the updated pdf).
> 6. *Baselines Configuration*: Thank you for raising this point. You are correct that we used IBL hyperparameters tuned for Neuropixels data without adapting them to calcium imaging, which may disadvantage the baseline. We are currently running a hyperparameter search for the IBL model on our calcium imaging data and will report the optimized results and full hyperparameter details for all baselines in the revised manuscript. This will ensure a fair comparison and help readers interpret the performance differences accurately.
> 7. *NDT3 citation*:  Thank you for pointing this out, we fixed it so that it now correctly states ”motor cortex microelectrode data from monkeys and humans”.
>
> References:
> [1] Li et al., 2024 (TMLR): https://openreview.net/forum?id=qHZs2p4ZD4
> [2] Azabou et al., 2023 (NeurIPS), https://arxiv.org/abs/2310.16046
> [3] Azabou et al., 2025 (ICLR) https://openreview.net/forum?id=IuU0wcO0mo
> [4] Wang et al., 2025 (Nature) https://www.nature.com/articles/s41586-025-08829-y
> [5] Zhang et al., 2024 (NeurIPS), https://arxiv.org/abs/2407.14668
> [6] Zhang et al., 2025 (ICML), https://arxiv.org/abs/2504.08201

---

> ### Author Response · Authors · 2025-12-03
> **MtM hyperparameter tuning**
>
> We have now completed an extensive hyperparameter sweep for the MtM model. We ran a sweep over transformer architecture parameters, learning rate, weight decay, and population masking ratio. Code and sweep values can be found here: https://anonymous.4open.science/r/IBL_MtM_model-wandb-sweep-experanto/README.md. These changes did not alter the causal-setting results (i.e. our forecasting evaluation). We attribute this to (a) the transformer being already sufficiently expressive, and (b) MtM’s masking ratios being fixed per regime - https://github.com/colehurwitz/IBL_MtM_model/blob/main/src/trainer/base.py#L134-L146. In particular, causal masking is always 60% of future responses masked (which for our data corresponds to 24/60 samples visible), which is nearly identical to the evaluation setting that we employ (25/60 visible). We thus kept this masking parameter. Similarly, adjusting the population masking ratio did not improve the forecasting metrics. Ultimately we report a forecasting metric of 0.18 pearson correlation for OmniMouse , compared to MtM’s 0.12. (the 0.2 for forecasting we report for an Omnimouse with 8 mice also had additional behavioural input, 0.18 is only with neuronal input).
>
> Next, we also tuned MtM for the population context evaluation that we report in our paper. For this metric, 1024 neurons are unmasked, predicting a fixed heldout set of neurons. Using a more aggressive population mask within the MtM population regime, we obtained 0.21 correlation compared to OmniMouse-5M’s 0.34. We emphasize again that OmniMouse is trained and evaluated on exactly the same data and the same 1024 neurons are given as context. We also note that both models are compared strictly after pretraining, with no finetuning.
>
> We believe that the fixed masking percentages per masking regime in MtM is the reason why the model performs worse than OmniMouse in the forecasting and population prediction tasks. In contrast, OmniMouse benefits from a broad diversity of masks—an ingredient we believe is central to its generalization and performance advantages, in addition to the multi-modal training with visual input.

---

### Author Response · Authors · 2025-11-27
**General Response to All Reviewers**

We thank all reviewers for their thoughtful engagement with our work. We are encouraged that reviewers recognized the value of our scaling analysis (**SAAW**), our *"timely contribution by systematically studying scaling laws"* (**Cuf1**), and the importance of our finding that neural modeling is data-limited (**Lnv7**, **8RvB**). Below, we summarize our main contributions and address the key concerns raised across reviews.

## **Statement of contribution**
Our primary contribution is a **systematic scaling laws analysis for neuronal data**. We scale transformers using established recipes and systematically investigate how model size and dataset size impact neuronal encoding and behavioral decoding performance. Our key findings are:
- Performance improves systematically with more data
- Performance saturates with model size beyond moderate scales
- We thus conclude that data, not model size, is currently the bottleneck for predictive accuracy in neural modeling

This last finding provides a clear directive for the field: progress requires larger and more diverse neural datasets.


In addition, we combine existing, proven architectural elements and training regimes into  a novel multi-modal multi-task model architecture with new capabilities (see point 3 below). We have now strengthened our baseline comparisons with data-matched experiments and an extensive POYO+ [4] hyperparameter search. These new results confirm that OmniMouse achieves state-of-the-art performance across nearly all tasks independent of data scale advantages.

## **Summary of reviewer discussions**

  In the following, we address the key concerns raised across reviews and provide our responses

### 1. **Baseline Comparisons**

Several reviewers raised concerns about baseline comparisons, particularly regarding baselines trained on fewer sessions (**Lnv7**) and missing POYO+ baseline (**SAAW**, **8RvB**). We fully agree this is important for interpreting our results, and **we have now conducted data-matched comparisons that we believe resolve this concern**.

**We now added POYO+ as a baseline.** We tuned it with an extensive hyperparameter search (see reply to **SAAW**).
We also extensively tuned MtM for the forecasting and the population context task (see reply to Cole Hurwitz). For the population context task, we chose the best condition for MtM (n=1024 neurons visible) to compare against OmniMouse.

**When trained on identical data, OmniMouse outperforms all strong specialized baselines across nearly all tasks**,
which shows the strength of our architecture independent of data scale advantages.

For the Sensorium 2023 benchmark, we trained OmniMouse on the same 10-mouse dataset provided by the competition. We note that while the data quantity is matched, our framework enables training across video boundaries— which was not done by previous models. We find that even without ensembling, our model outperforms the competition winner on both the main track and the out-of-distribution track.
We summarize all results in the table below.

---
**Tab.1: Baseline comparison**
|Model|Fcst|Fcst+S|Pop|Pop+S|Gaze|Pupil|Running
|:-|:-:|:-:|:-:|:-:|:-:|:-:|:-:|
|**MtM** (Zhang et al., 2024) [1]|0.12|✘ |0.21|✘|✘|✘|✘
|**Latent Model** (Schmidt et al., 2025) [2]|✘|0.18|✘|0.16|✘|✘|✘
|**CEBRA** (Schneider et al., 2023) [3]|✘ |✘|✘|✘|0.52|0.55|**0.51**
|**POYO+** (Azabou et al., 2025) [4]| ✘|✘|✘|✘|0.56|0.63 |0.47
|**OmniMouse-5M (data-matched)**|**0.18**|**0.30**|**0.34**|**0.27**|**0.68**|**0.66** |0.44
|||||||||
|**OmniMouse-1M (full data)**|0.18|0.33|0.36|0.35|0.75|0.73|0.55
|**OmniMouse-5M (full data)**|0.22|0.34|0.37|0.35 |0.76|0.74|0.57
| **OmniMouse-20M (full data)**| 0.23| 0.35 | 0.38|**0.37**|0.78 | 0.75| 0.73
|**OmniMouse-80M (full data)**|**0.25**|**0.36**|**0.39**|**0.37**|**0.80**|**0.76**|**0.75**
|**OmniMouse-300M (full data)**|**0.25**|**0.36**| **0.39**|**0.37**|**0.80**|**0.76**|0.73

Results displayed in **bold** indicate the highest score per task in either the data-matched condition (8 sessions; first 5 rows) or when using the full dataset (323 sessions; last 5 rows).
Conditions: Forecasting (*Fcst*), forecasting + stimulus (*Fcst+S*), population context (*Pop*), population context + stimulus (*Pop+S*). Behavioral decoding: Pupil location (*Gaze*), pupil size (*Pupil*), running speed (*running*).

---



**Tab.2: Sensorium 2023 benchmark results**
  |Model|Training|Main track ↑|OOD track ↑|
  |:-|:-|:-|:-|
  |DwiseNeuro-$\Sigma$ [5] |end-to-end|0.291|0.221
  |||||||||
  |OmniMouse-80M| frozen |0.313 $\pm$ .001 |0.274$\pm$ .001
  |OmniMouse-80M-$\Sigma$| frozen |0.327|0.288
  |OmniMouse-5M-Unimodal|end-to-end|0.288$\pm$.003|0.256 $\pm$.002
  |OmniMouse-5M-Unimodal-$\Sigma$|end-to-end|0.332|0.296
  |OmniMouse-5M|end-to-end|0.295 $\pm$ .005|0.263 $\pm$ .003
  |OmniMouse-5M-$\Sigma$|end-to-end|0.327|0.293

Note that end-to-end trained OmniMouse models are data-matched to DwiseNeuro [5], the previous state-of-the-art model.

---

> ### Author Response · Authors · 2025-11-27
> **Continued general response to all reviewers**
>
> ### 2. **Code, Data, and Reproducibility**
>
> We apologize for the oversight in not including availability statements. We have released:
>
> - Complete model code (training, evaluation, fine-tuning, and inference): https://tinyurl.com/4s3nyksr
> - Full data-loading and processing code: https://tinyurl.com/3tx9cebj
>
> The full dataset—the largest of its kind with naturalistic visual stimulation, behavior, and visual cortex recordings—will be publicly released within six months.
>
>
> ### 3. **Clarifying our architectural contributions relative to prior work**
>
> We appreciate the opportunity to clarify how OmniMouse differs from POYO(+) [1, 2], NEDS [3] and other prior work. While we build upon proven components, our contribution lies in integrating and extending them into a unified framework that handles arbitrary combinations of neural forecasting, prediction of sub-populations, stimulus encoding, and behavioral decoding through flexible masking.
>
> - **Single-neuron tokenization with masking semantics**: We adopt POYO+/POCO-style [5] single-neuron tokenization for calcium traces, extended to the decoder. This enables per-neuron masking on both input and output sides with high temporal precision—simply by removing tokens for masked neurons. In contrast, POYO does not predict neuronal responses, and NEDS tokenizes at the population level, making it difficult to distinguish zero-as-no-activity from zero-as-masked.
>
> - **Hierarchical video encoding**: We use a lightweight, hierarchical vision transformer with local-attention layers [6] to tokenize video at frame-level granularity, allowing variable temporal masking of visual context (e.g., predicting neural responses from partial stimulus sequences). These video features fuse with neural and behavioral embeddings through our transformer stack, creating a unified multi-modal representation. Neither POYO/POYO+ nor NEDS incorporate visual input.
>
> - **Structured masking framework**: Inspired by MtM [4], we designed 119 masking configurations spanning our core tasks as well as partial masking scenarios (e.g., partial video + partial neural population). Training across this diverse masking space allows seamless task switching purely through mask configuration and enables the model to generalize to arbitrary input combinations at test time.
>
> To clarify the capabilities of each approach, we have compiled a comparison table showing which tasks (masking, forecasting, encoding, decoding) and conditioning configurations are supported by OmniMouse and prior work. We include this below for reference.
>
>  ---
> **Tab.3: Comparison of multi-modal multi-task capabilities of OmniMouse and prior work**
> Output|Conditioned on|OmniMouse| POYO+|MtM|NEDS|POCO|Wang et al.|Schmidt et al.|
> |:-------|:---------------|:---------:|:-----:|:---:|:----:|:----:|:----:|:-------:|
> |***Population context***| | | | | | | | |
> | Held-out neurons | Neural activity | ✔️ | ✘ | ✔️ | ✘ | ✔️ | ✘ | ✘
> | Held-out neurons | Neural activity + Stimulus | ✔️ | ✘ | ✘ | ✘ | ✘ | ✘ | ✔️
> | Held-out neurons | Neural activity + Behavior | ✔️ | ✘ | ✘  | ✘ | ✘ | ✘ | ✘
> | Held-out neurons | Neural activity + Stimulus + Behavior | ✔️ | ✘ | ✘ | ✘ | ✘ | ✘ | ✘
> | ***Forecasting*** | | | | | | | | |
> | Future activity | Neural activity | ✔️ | ✘ | ✔️ | ✔️ | ✔️ | ✘ | ✘
> | Future activity | Neural activity + Stimulus | ✔️ | ✘ | ✘ | ✘ | ✘ | ✘ | ✔️
> | Future activity | Neural activity + Behavior | ✔️ | ✘|  ✘ | ✔️ | ✘ | ✘ | ✘
> | Future activity | Neural activity + Stimulus + Behavior | ✔️ | ✘ | ✘ | ✘ | ✘ | ✘ | ✘
> | ***Encoding*** | | | | | | | | |
> | Neural activity|Stimulus|✔️| ✘ | ✘ | ✘ | ✘ | ✘ | ✘ |
> |Neural activity| Behavior | ✔️ | ✘ | ✘ | ✔️ | ✘ | ✘ | ✘ |
> |Neural activity | Stimulus + Behavior | ✔️ | ✘ | ✘ | ✘ | ✘ | ✔️ | ✘
> |***Decoding***| | | | | | | | |
> |Behavior| Stimulus |✔️| ✘ | ✘ | ✘ | ✘ | ✘ | ✘ |
> |Behavior|Neural activity |✔️| ✔️ | ✘\* | ✔️ | ✘ | ✘ | ✘
> |Behavior|Neural activity + Stimulus | ✔️ | ✘ | ✘ | ✘ | ✘ | ✘ | ✘
> |Stimulus|Neural activity  | ✘ | ✔️ | ✘ | ✔️ | ✘ | ✘ | ✘
>
> \* Behavioral decoding not included in multi-modal training.
>
> ---
>
>
> ### **Concluding Remarks**
>
> We believe these contributions—validated by our new data-matched experiments—represent a meaningful advance *toward* multi-modal multi-task foundation models of the brain. We thank the reviewers for helping us strengthen the paper and we welcome any follow-up discussions.
>
>
> References:
> [1] Azabou et al., (POYO), 2023, https://arxiv.org/abs/2310.16046
> [2] Azabou et al., (POYO+), 2025 https://openreview.net/forum?id=IuU0wcO0mo
> [3] Zhang et al., 2025 (NEDS), https://arxiv.org/abs/2504.08201
> [4] Zhang et al., 2024, (MtM), https://arxiv.org/abs/2407.14668
> [5] Duan et al., 2025, (POCO)  https://arxiv.org/abs/2506.14957
> [6] Ryali et al. 2023, https://arxiv.org/abs/2306.00989
> [7] Wang et al. 2025, doi: 10.1038/s41586-025-08829-y
> [8] Schmidt et al., 2025, https://arxiv.org/abs/2410.16136

---

### Author Response · Authors · 2025-12-04
**Author Rebuttal Summary**

We thank the reviewers for their thoughtful engagement with our work. Our paper introduces OmniMouse, a novel multi-modal multi-task model for neuronal data, and presents a systematic scaling analysis on the largest single-neuron dataset of its kind.


**Addressing Reviewer Concerns**

The main criticisms raised by reviewers centered on (1) baseline comparisons, particularly whether our improvements stem from architectural and masking advances or simply more training data, (2) missing comparisons to recent transformer models such as POYO+, (3) clarity of our architectural contributions relative to prior work, and (4) code availability.

We have addressed each of these concerns as follows (please see the general response to all reviewers below for details):


- *Baseline comparisons and data-matching* (**Lnv7**, **SAAW**, **8RvB**): We conducted data-matched experiments where OmniMouse and all baselines are trained on identical data. These experiments confirm that OmniMouse achieves superior performance independent of data scale advantages, isolating our architectural+training contributions.

- *Extensive hyperparameter tuning for MtM and POYO+* (**SAAW**, **8RvB**): We performed comprehensive hyperparameter sweeps for both MtM and POYO+. Even after tuning, OmniMouse-5M (data-matched) outperforms both baselines across nearly all tasks (see Tab. 1 in general rebuttal).

- *State-of-the-art performance*: OmniMouse establishes state-of-the-art in the Sensorium 2023 benchmark where our model outperforms the competition winner on both the main track and out-of-distribution track—without ensembling and when trained on the same data (Tab. 2).

- *Clarified architectural contributions* (**Cuf1**, **SAAW**): We now provide a detailed comparison table (Tab. 3) illustrating how OmniMouse's unified framework—combining single-neuron tokenization with masking semantics, video encoding, and a structured masking framework—enables capabilities not supported by prior work (POYO+, MtM, NEDS, POCO).

- *Code release*: We have released complete code for training, evaluation, fine-tuning, inference, and data processing. The full dataset will be publicly released within six months.


**Manuscript Revisions**


In light of reviewer feedback, we have made the following revisions to strengthen the manuscript:
- Baseline comparisons (table 2): we have extended the baseline comparison with data-matched results across all tasks and added new baselines models.
- Added table 3: detailed Sensorium 2023 benchmark comparison demonstrating state-of-the-art on main and OOD tracks, with and without ensembling
- Clarified state-of-the-art performance claims to distinguish architectural and training contributions from data scale advantages
- Added Suppl. Fig. S1: Qualitative visualization of model predictions
- Added Suppl. Fig. S3: Systematic evaluation across mask configurations demonstrating generalization to novel input combinations at test time
- Added missing citations and clarified the relationship to previous related work
- Fixed typos throughout the manuscript


Please refer to the detailed rebuttal summary below and individual reviewer responses for further information.


We sincerely thank the reviewers for their thoughtful critiques and engagement, which have strengthened both our empirical evaluation and the clarity of our contributions.

---

### Meta-Review · Area_Chair_tbYt · 2025-12-24

**Summary:**

The four reviewers assigned ratings of 6, 6, 6, and 4, with corresponding confidence levels of 3, 3, 3, and 5. Nearly all concerns raised by the reviewers have been thoroughly addressed in the authors’ rebuttal, and the overall scores satisfy the conference’s acceptance threshold. Specifically, the paper’s primary contribution is its constructed dataset: OmniMouse, which contains 3.3 million neurons from the visual cortex of 78 mice across 323 recording sessions, with the authors committing to make the full dataset publicly available within six months. While the proposed model architecture has limited technical novelty, this study pioneers the systematic application of scaling laws methodology to a single-neuron, multi-modal brain foundation model. I tend to recommend acceptance.

**Reviewer Concerns:**

1. Addressed Reviewer SAAW’s concerns: added POYO+ benchmark and shared full data-loading code plus multi-modal model code, and they commit "The full dataset will also be released within the next six months."

2. Addressed Reviewer Lnv7’s concerns: Added OmniMouse-5M trained on the same 8-mouse dataset as baselines (outperforming most baselines) and validated on Sensorium 2023; clarified the model’s scope and proposed future neuron-agnostic tokenization strategies.

Outstanding concerns from Reviewer Lnv7: No implementation/validation of neuron-agnostic tokenization and per-neuron embeddings still limit zero-shot generalization to unseen neurons/animals.

3. Addressed Reviewer Cuf1’s concerns: Clarified novel architectural components; added qualitative visualization; conducted modality disentanglement analysis; explained low correlation metrics and data-limited regime claim; responded to visual stimulus-pattern attribution questions with MEI framework references and discussion additions.


4. Addressed Reviewer 8RvB’s concerns: Moved data upsampling notes to main text, added neuron-hours for dataset sizing, clarified sampling novelty and revised wording; supplemented POYO+ baseline comparison with hyperparameter tuning results; fixed all presentation typos and omissions; confirmed dataset release in 6 months and shared code; clarified checkpoint training and 40 behavior frames rationale.

**Reviewer Scores:**

The four reviewers assigned ratings of 6, 6, 6, and 4, with corresponding confidence levels of 3, 3, 3, and 5.

---

### Decision · Program_Chairs · 2026-01-26

Accept (Poster)